# The global extent of the grassland biome and implications for the terrestrial carbon sink

Land cover data are commonly used to model the terrestrial carbon (C) sink, yet these data have wide margins of error that significantly alter estimates of global C storage. Here we demonstrate this data vulnerability in grasslands, which are critical to C cycling but whose estimated distribution has varied by >50 million $km^2$ (3.5–42% of the Earth's terrestrial surface). Comparing multiple high-resolution land cover products with expertly annotated grassland data from six continents, we show sources of mapping error and discuss C implications based on 2023 United Nations (UN) FAO estimates. Past misidentification arose from inconsistent definitions on grassland identity and classification flaws especially relating to woody plant cover. Correcting these errors adjusted grassland coverage to 22.8% of the terrestrial land base (30.1 million $km^2$), elevating UN projections of soil C stocks to 155.02 Pg (0–30 cm depth). These findings underscore the challenges of biome mapping for ecosystem accounting and policy, when lacking field-validated remotely sensed data.

High-resolution spatial mapping of land cover is increasingly used to quantify indicators of global sustainability including climate solutions[1–5]. Recently developed high-resolution land use/land cover (LULC) products such as World Cover (WC) by the European Space Agency (ESA), Environmental Systems Research Institute (ESRI)'s Land Cover (LC) and Dynamic World (DW) generated by Google help improve long-standing limitations of mapping the spatial extent of global land cover, with consistent and relatively affordable data available for all regions of the Earth[6–10]. While providing a substantially improved characterization of the Earth's land surface, these products still possess large classification error despite their ability to map land features at 10 m resolution[11–15]. The main source of error centres on the absence of systematic fine-scale expert knowledge necessary for validation, with image interpretation typically conducted by remote sensing experts lacking local knowledge of the terrestrial features that they are attempting to map[11,16,17]. Although fine-scale error is increasingly being minimized, inconsistencies of even a few metres quickly compound when modelling global-scale processes (Supplementary Fig. 1). This is especially problematic for quantification of the terrestrial carbon (C) sink[16], as discrepancies in estimates of land cover bias model outcomes for C stocks and fluxes[16,17]. Given the importance of global C modelling for quantifying ecosystem-based 'Natural Solutions' to achieve

C neutrality by 2050 (for example, 'per area' estimation of C uptake and storage[18,19]), we need to better understand the sources of error in remotely sensed estimates of Earth-system processes[14].

These challenges are especially pronounced in global grasslands. Grasslands are critical for all facets of human sustainability given their vast extent, importance for food production, direct support of the livelihood of ~850 million people, vulnerability to anthropogenic disturbances and provision of habitat for biodiversity including large numbers of endemic plants and many of the world's at-risk megafauna[17,20,21]. Grasslands are also critical for global C dynamics, as their soils are thought to contain ~20% of global C while contributing ~18% of the yearly total for the terrestrial C sink (0.5 Pg C stored per year[17]). However, these C contributions can only be viewed as broad approximations because published estimates of total grassland area vary by tens of millions of $km^2$ (Table 1). These inconsistencies derive from a range of sources, including variability in how grasslands are defined, potential changes in global grassland area over the past several decades and challenges in using remote images to accurately differentiate grassland from other land cover types in the absence of local expertise[17,22].

Such wide differences in estimated area matter for modelling the contribution of grasslands to global C cycling. In the UN's Food and Agriculture Organization (FAO) 2023 report on global grassland storage

✉e-mail: asm@uoguelph.ca; matthias.siewert@umu.se

**Table 1 | Published estimates of global grassland coverage, showing wide variation in total area**

| Authors | Year | % global coverage | Km² (million) | Source |
|---|---|---|---|---|
| Olson et al.[53] | 1983 | 42.8 | 55.5 | Original[a] |
| Bai & Cotrufo[54] | 2022 | 40.5 | 52.5 | #White et al.[27] |
| Suttie et al.[55] | 2005 | 40.5 | 52.5 | #White et al.[27] |
| White et al.[14] | 2000 | 40.5 | 53.5 | GLCCD 1998[56] |
| Bardgett et al.[49] | 2021 | ~40 | - | #White et al.[27] |
| Sun et al.[57] | 2022 | ~40 | - | #White et al.[27] |
| O'Mara[58] | 2012 | 37 | 50 | #Loveland et al.[28] |
| Chang et al.[59] | 2021 | 37 | 48.1 | IPCC 2019[60] |
| Loveland et al.[28] | 2000 | 35 | 51.3 | GLCCD 1998[56] |
| Whittaker & Likens[61] | 1973 | - | 40 | Original[b] |
| Goldewijk et al.[62] | 2007 | - | 33.4 | Original[c] |
| Latham et al.[63] | 2014 | 31.5 | - | Original[d] |
| Arneth et al.[64] | 2019 | 30 | - | FAO 2018[65] |
| Lal[35] | 2004 | 29.4 | 44.5 | IPCC 2000[66] |
| Schellburg et al.[67] | 2008 | 26 | 34.4 | FAO 2008[68] |
| Liu et al.[69] | 2019 | ~25 | - | #Lieth 1978[70] |
| Lauenroth[71] | 1979 | 25 | 33 | Shantz 1954[72] |
| Lieth[70] | 1975 | 24.3 | - | Original[e] |
| Shantz[72] | 1954 | 24 | 20.3 | Original[f] |
| Xia et al.[73] | 2014 | ~20 | - | #Lieth 1978[70] |
| Scurlock & Hall[31] | 1998 | ~20 | 22 | #Lieth 1978[70] |
| ESA WC | 2020 | 24.3 | 32.1 | Original |
| Google DW | 2020 | 5.0 | 7.4 | Original |
| ESRI LC | 2020 | 3.5 | 4.6 | Original |

#Indirect citation (citing a paper that cites an original derivation of grassland extent). [a]Merges potential and existing vegetation maps: grassland totals pool grassland, tundra, pasture and savanna. [b]Merges terrestrial maps including Vahl's climate and vegetation zones (1949), for savanna, grassland, tundra and shrubland. [c]Merged maps of ref. 28; Bartholomé et al.[74]. [d]GLC-SHARE: synthesis of the existing global information sources into a single database. [e]Merges potential and existing vegetation maps. [f]Merging of 'best maps available', excluding tundra and shrubland (44.4 million km² with those added). These differences include both percent coverage (%) of the Earth's terrestrial ice-free surface and total grassland areas, although this latter total is not always mentioned (millions of km²). Divergence in estimated totals tends to derive from whether tree cover, shrub cover and/or tundra are included as 'grassland' (light grey, >30% global coverage) or if estimates focus solely on 'open grassland' (dark grey, <30%). Variation within these two groupings can also be sizeable, with every 1% difference in land cover equivalent to ~1.3 million km². '~' refers to estimates where the authors acknowledge uncertainty, using terms such as 'up to', 'over' or 'approximately', without giving an exact total.

of carbon[17], grasslands were estimated to store 63.5 Mt C in 2010 based on a land cover estimate of 17.9 million km² (CCI_LC [Climate Change Initiative_Land Cover]; Envisat satellite, Supplementary Table 1). Yet in that same year of 2010, a different estimate derived using the Moderate Resolution Imaging Spectroradiometer (MODIS) on board the TERRA and AQUA satellite constellation, calculated grassland coverage at 30.5 million km² (Supplementary Table 1). Wide differences also occur among WC, LC and DW, reflecting issues with how grasslands are defined and classified remotely (Table 1). If global land cover mapping is to become a dependable and authoritative source to quantify large-scale environmental, economic and social indicators of sustainability, including those derived from grasslands, then such measurement shortcomings must be better addressed[11,14,23].

Here we use high-resolution (10 m) globally distributed grassland data from 504 sites on 6 continents (Fig. 1) to illustrate these issues. We do this by cross-referencing 387,600 expert-validated pixels from Google and Bing satellite imagery with WC, LC and DW estimates of grassland coverage. Our comparisons allow us to isolate sources of overestimation and underestimation error by the three land cover products, approximate the most likely extent of the grassland biome and explore the implications of this updated approximation for grassland impacts on C stocks and fluxes in relation to the most recent UN FAO estimates[17]. The power of our analysis derives from our high-resolution data validated by 157 grassland researchers in 60 countries, creating

a 'reference classification'[14,15] against which the performance of WC, LC and DW can be assessed (Supplementary Fig. 2). This reference classification derives from a definition of grassland extending from ≥5% grassland vegetation in barrenland to forested grassland with ≤75% woody plant cover, thereby capturing various grassland cover types including savanna, shrubland, planted pasture and tundra (see Methods). Our comparative approach mirrors the validation protocols used by the world's leading land cover products, where 1,000 s of sampling units are visually assessed to test effectiveness and train mapping protocols[6–13]. Where our work differs is the direct on-the-ground familiarity of our sites and their surrounding area, allowing us to differentiate grasslands even among fuzzy or cryptic coverage in ways otherwise not possible (Supplementary Fig. 1). WC, for example, uses annotated pixels from 141,000 locations globally extending across all biomes, not just grasslands, that are validated by visually assessing pixel identity but without local knowledge[7,8,11]. DW and LC use near-identical training and validation protocols based on 24,000 Sentinel 2 tiles and 5 billion pixels, but again without field-verified accuracy[10,11]. As we will show, this lack of local knowledge results in significant error (Supplementary Fig. 1). Grasslands are especially difficult to remotely classify because of their highly variable spectral signals relating to phenological, edaphic and herbivore-related influences on biomass, as well as challenges identifying ground cover under canopies of woody plants or in sparsely vegetated barren lands. As a result, remotely

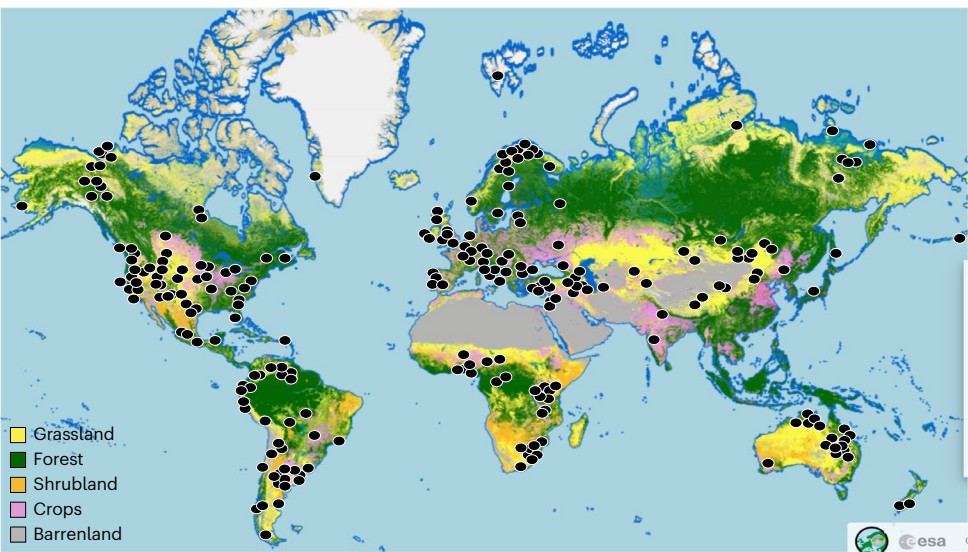

**Fig. 1 | Grassland sites of this study, in relation to the WC land cover map.** Most sites (indicated by the black dots) contain 11 reference grids (with some exceptions, see Methods) with each grid composed of 100 10 m × 10 m pixels (Supplementary Fig. 2). We classified each pixel (387,600 pixels in total) into 1 of 7 grassland types and 9 non-grassland land use classes (for example, settlement, crops, forest; see Supplementary Table 3). We chose WC for this map, instead of LC or DW, because it was the most accurate land cover product for grasslands (see Supplementary Table 2). Photo credit: ESA WC project 2021[7].

sensed grassland distributions have had some of the lowest accuracies so far among mapped biomes[11].

In addition, we classified each of the 387,600 pixels into 1 of 16 cover types (Supplementary Table 2). Seven are forms of grassland, while the remaining 9 are non-grassland features such as settlements, crop fields and forest or shrubland lacking grassland understories. These 16 classifications, in turn, allow us to detect and quantify the underlying causal sources of estimation error for WC, LC and DW, which can take two forms: user's accuracy or 'false positives' where a pixel is incorrectly classified as grassland when it is actually a non-grassland cover type (also known as 'errors of commission' or Type I error), and producer's accuracy or 'false negatives' where a grassland pixel is mistakenly labelled as something else such as forest or crop field (also known as 'errors of omission' or Type II error). Our expert-validated data allow us to close the gap between remotely sensed and on-the-ground classification, clarifying the benefits and shortcomings of current high-resolution land cover products for modelling Earth-system processes. Our work targets the grassland biome but is illustrative of the benefits and pitfalls of remotely sensed ecosystem assessments generally.

## Results and discussion

Our high-resolution analysis of grassland coverage reveals that even the most accurate of remotely sensed land cover mapping possesses substantial error, with over- or underestimation of grassland causing misclassification of pixels at least 15% of the time depending on the type of error and which land cover product is being assessed (Fig. 2 and Supplementary Table 3). Indeed, the estimated extent of grassland among the three high-resolution land cover products differed by millions of km² (Table 1), despite all deriving from Sentinel-based satellite imagery (see Methods). Such discrepancies reveal the sensitivity of land cover estimates to how biome features are defined and classified with satellite-derived data. They also indicate that estimates of the grassland C sink will differ widely depending on which product is utilized. In total, this work demonstrates that attempts to remotely generate sustainability estimates must be clear on the high level of potential error that can be involved and the irreplaceable importance of local field data to improve accuracy no matter how detailed the land cover platforms may be.

As stated, product accuracy varied widely and was sensitive to different forms of error. WC was more accurate for grassland mapping

than DW and LC, correctly identifying 240,266 of 281,735 pixels globally (80.3% user's accuracy (false positives), 85.3% producer's accuracy (false negatives); Supplementary Table 3). Yet, WC was more likely to misidentify non-grassland by labelling a location as grassland when it was not (Supplementary Table 3). WC only correctly identified 59,064 of 105,865 non-grassland pixels (53% user's accuracy, 44.2% producer's accuracy). As a result, the combined WC accuracy for grassland and non-grassland features was 74.1%. This inability of WC to correctly classify ~25% of all pixels illustrates the high level of error associated with even the best remotely sensed land cover products. For LC and DW, their overall accuracy was even lower with mislabelling of 60% and 57% of all pixels, respectively.

LC and DW were especially poor at detecting grassland. DW only correctly identified 70,864 grassland pixels, with grassland often mistakenly labelled as a non-grassland attribute such as forest (86.8% user's accuracy, 25.2% producer's accuracy). DW performed much better for non-grassland features, correctly identifying 95,130 of 105,865 pixels, with most errors being false positives (31.1% user's accuracy and 89.1% producer's accuracy). LC had the lowest performance for grassland, only correctly identifying 58,386 pixels, with most error deriving from false negatives (87% user's accuracy, 20.7% producer's accuracy). As with DW, its classification accuracy for non-grassland features was much better (30.3% user's accuracy, 91.8% producer's accuracy). The tight correlation in performance by LC and DW (Supplementary Figs. 3 and 4) reflects their similar training data and classification schemes[5] (Supplementary Tables 6 and 5). We see this correlation when examining the sensitivity of the three cover products to different environmental parameters (Supplementary Figs. 3 and 4; see Methods). For example, the performance of LC and DW was positively associated with sites with higher annual precipitation (Supplementary Fig. 3). This may be explained by the high accuracy of both products for detecting planted pasture (Supplementary Table 3) compared to other grassland types – planted pastures can be widespread in temperate regions with higher rainfall that might otherwise support forest.

Taken collectively, the sources of error for estimating grassland extent derived from two factors: the definitions of grassland ('definition error') and the difficulty in remotely resolving certain cover types as represented by the user's and producer's accuracies ('classification error'). Definition error occurred when areas that support grassland were excluded. This happened for logistical reasons, such as WC

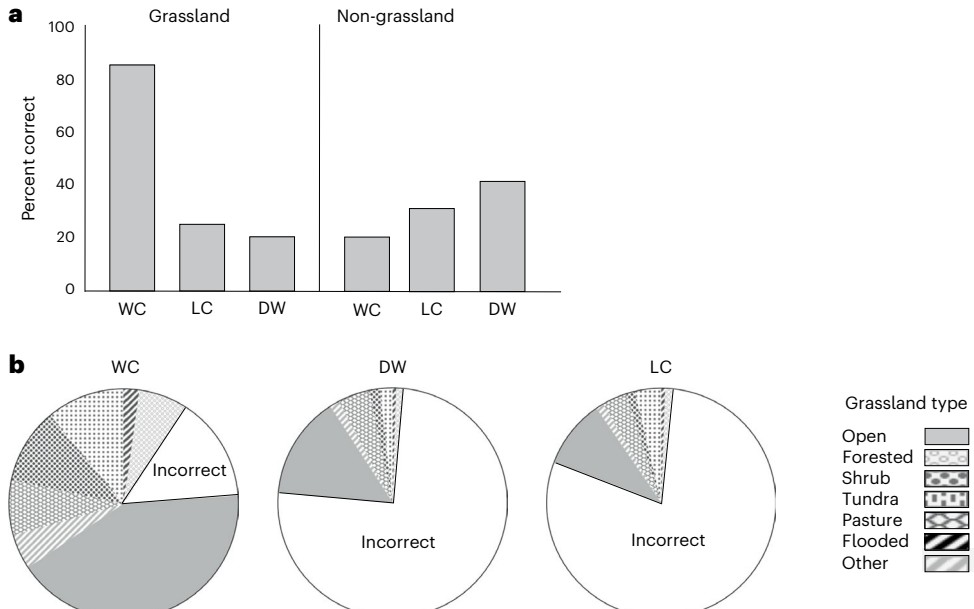

**Fig. 2 | Differences in accuracy for identifying grassland and non-grassland land cover in 10 m × 10 m pixels among three high-resolution land cover products. a**, Percent differences in accuracy between the overall categories of grassland and non-grassland cover. WC correctly identified grassland in 85% of 281,735 pixels but was the least accurate for non-grassland features such as crop field and wetland. **b**, Number of correctly identified pixels per seven classes of grassland, as well as pixels incorrectly identified. Open grassland was the most frequent cover type to be correctly identified by all products, followed by tundra for WC and LC, and planted pasture for DW. The numbers of incorrectly identified pixels are significantly higher for DW and LC than for WC (see Supplementary Table 2).

defining grasslands as areas with <10% tree canopy given the difficulty of remotely observing grassland understories below forest cover. This decision resulted in the exclusion of grasslands with substantial tree cover but a grassland understory (that is, 'savanna'), including in tropical, temperate and sub-Arctic regions (Supplementary Fig. 1). There was also some suggestion of management-based sources of definition error relating to the delineation of global 'rangelands'. Rangelands, often defined as areas that support domesticated or native livestock, are critically important for global food production and are often viewed as synonymous with grassland. However, grazing animals are not always confined to grassland, with many shrublands and desert areas also being used; as such, rangelands have a larger distribution than grasslands globally[24–26], with an estimated area of at least 34 million km². Recognizing this, LC recently merged 'grassland' with 'shrubland' to create a broader category of 'rangeland' (see Supplementary Fig. 1), which they assessed as covering 37% of the Earth's ice-free surface. This shift now means that LC no longer explicitly maps the occurrence of 'grasslands'.

The other major source of land cover error derived from classification inaccuracies, where one or more of the land cover products incorrectly classified pixel identity. Such errors can relate in part to how grassland was defined but also occur due to technical issues relating to spectral challenges associated with grassland detection from space. For example, we classified 30,166 pixels as 'wooded grasslands', which WC only identified as grassland 65% of the time, with the remainder being incorrectly listed as 'forest' or 'shrubland' (Supplementary Table 3). Cropland caused challenges, especially harvested grain fields (sometimes incorrectly classified as grassland) or long-established planted pasture (sometimes incorrectly classified as crop field, despite not being cultivated annually). Similarly, we were able to separate grasslands in settlements from uncultivated grassland, which also caused confusion for LULC mapping. For example, WC incorrectly labelled 10,400 of 17,533 heavily managed 'artificial' pixels as 'grassland' when these tended to be lawns in residential areas.

We used these assessments of mapping accuracy, combining both over- and underestimation error for each land cover product, to approximate a corrected total extent of grassland (see Methods). In the absence of this correction, the coverage of grassland for WC, LC and DW diverged substantially from 3.5%–24.4% (a gap of 27.5 million km²) of the Earth's ice-free terrestrial surface estimated at 131,319,290 km² (Supplementary Table 5). After correcting for estimation error, these totals converged somewhat especially for LC and DW. WC has an adjusted total grassland coverage of 30.1 million km² (22.8% of the terrestrial land surface), with LC at 19.5 million km² (14.2%) and DW at 25.6 million km² (17.2%). Given the much higher user and producer accuracy for WC, especially its ability to correctly identify actual grassland, we suggest that its adjusted total of 30.1 million km² is the most accurate estimate of global grassland area especially since this total corrects for WC's inaccuracy at detecting savanna.

This corrected grassland total for WC is substantially lower than commonly reported (Table 1), including two of the most cited estimates: ref. 27 (40.5% global cover) and ref. 28 (35%) (Supplementary Table 4). Although grassland loss in some global regions since 2000 could contribute to the larger estimates at that time, the more likely cause derives from changes in the definition and classification of remotely sensed grassland images. Indeed, both refs. 27,28 used much coarser-resolution data than available today (advanced very-high-resolution radiometer (AVHRR) satellite data with a 1 km resolution). In terms of definition error, ref. 27 combined grassland and shrubland together into a single 'grassland' category, which would overinflate grassland occurrence because not all shrublands have grassland understories. Similarly, ref. 28 excluded tundra from their 'grassland' category, which probably explains why their estimate is 5% smaller than that of ref. 27, given that tundra covers ~5–7% of the Earth's surface (Supplementary Table 4). As for higher coverage in 2000 being explained by subsequent grassland loss, both CCI_LC (1992–2020) and MODIS (2001–2022) show that grassland totals have increased over the past several decades (+1.5% and +6.8% respectively; Supplementary Table 1), which we assume to relate to factors such forest clearance for pasture or farmland abandonment. Nevertheless, these two products differ substantially in estimating total area, with MODIS having an average of 51% more grassland than

CCI between 2001–2022, representing a gap of at least 11 million km$^2$ (Supplementary Table 1). We see these same large discrepancies in current high-resolution WC, LC and DW land cover products, despite their much higher spatial precision compared with earlier satellite imagery (Supplementary Table 1). In total, these issues illustrate the challenges that have inhibited the accurate reporting of grassland distribution. Unfortunately, an on-going reliance on older estimates tends to inadvertently reinforce, and sometimes conflate, the reporting of grassland coverage (that is, the 'scientific telephone' effect[29]). Some estimates have attempted to reduce this risk by being highly conservative (see Table 1), but these are not necessarily better because they tend to strictly emphasize open grassland while excluding savanna or tundra.

In turn, the wide spectrum of cited grassland cover estimates (Table 1) biases the modelling of global C stocks and their expected sequestration potential. In short, C stocks derive from the combination of area (for example, Gt C km$^{-2}$) and volume (for example, g C cm$^{-3}$ of soil), with both shaped by contextual factors such as climate, soil nutrients and past land use that affect C losses and gains[16,30]. For area, a major challenge for projecting the global grassland C sink is that high-resolution mapping products differ so widely in their estimates. Indeed, the mislabelling of cover type will have large impacts on the spatial modelling of C as biomes tend to process and store C differently, especially forests versus grasslands, given their differences in annual primary production, nutrient cycling and fire frequency[30–34]. Because of these challenges, past estimates of grassland contributions to the terrestrial C pool need to be viewed with caution. In one of the most widely cited projections, ref. 35 relied on previously created biome maps to estimate grassland coverage at 44.5 million km$^2$ including tropical grassland and savanna, tundra, and temperate grassland and scrublands. On the basis of these totals, grassland soils were determined to support 32.04% of the world's soil organic carbon pool based on a grassland C pool of 552 billion tonnes to 1 m depth and a total global C pool size of 1.72 trillion tonnes[35]. Our analysis here suggests that 44.5 million km$^2$ is an overestimation, at least compared to our adjusted WC estimate of ~30.1 million km$^2$. We see the same sensitivity to per area estimates in the 2023 UN FAO report on grassland C storage[17], which relied on grassland performance in 2010 using CCI_LC land base projections for that year. Adjusting their projections of grassland C stocks at 51.5 t C ha$^{-1}$ to 30 cm depth with our corrected estimate increases the size of the soil C pool by 68% to 155.02 billion tonnes. Such adjustments are critical for better estimating the magnitude of the grassland C sink and for demonstrating the large role that grasslands can play in providing Natural Solutions to global C capture and storage until anthropogenic $CO_2$ emissions begin to drop.

There are several caveats to our analysis. Our assessment criteria were shaped by our own definition of grassland, which we set from ≥5% coverage of grassland vegetation to ≤75% canopy cover of trees or shrubs. Changes to either of these boundaries would alter our grassland totals, although the range we used represents commonly described parameters for defining 'grassland'[20,36]. It should be noted that our grassland definition is more encompassing than sometimes used to estimate grassland, especially our inclusion of woodland canopy cover, and yet we still found far lower totals than are commonly reported (Table 1). In addition, there was some level of regional clustering of our sites despite our efforts to sample all grassland areas globally (Fig. 1). Indeed, our sampling grids span all continents but more occur in North America (1,222 100 × 100 m grids) and Europe (968 grids), compared to 526 in Asia, 458 in South America, 361 in Africa and 209 in Oceania. We tested for spatial clustering using nearest-neighbour analysis, but found our sites to be spatially dispersed ($Z = 4.62$, $p < 0.0001$, nearest-neighbour ratio 1.13). Similarly, we investigated potential bioclimatic patterns of disagreement in our expert classification data but found none (for example, Supplementary Fig. 5). Finally, most past efforts to assess the accuracy of remote global biome mapping have used randomized approaches targeting all cover types[14,15].

Our approach was non-random as it strictly targeted grasslands. Further, our reliance on local expert knowledge meant that many locations supported some form of active research or management including conservation, thereby creating possible bias towards certain forms of grassland. We accounted for this by randomly assigning the locations of ~91% of all pixels within 5 km radius of most locations, which resulted in 27% of all pixels being non-grassland (see Methods).

In summary, our findings illustrate the shortcomings of using high-resolution remotely resolved products in the absence of corroborating field data, whether to assess biome distribution or infer benefit relating to ecosystem services and sustainability such as C accounting[3,11]. Our corrected estimate for grassland coverage of 22.8% of the terrestrial land surface or 30.1 million km$^2$ deviates from many widely cited totals, which are often much larger and continue to serve as integral components of global models for terrestrial C dynamics. As we show, coverage errors derive from a range of sources centering on how grasslands are defined and classified. Definitional choices such as excluding grassland with >10% woody plant cover are understandable as this reduces false positive error, given the challenges of remotely differentiating non-grassland forest versus heavily treed grassland savanna. However, this choice magnifies the misclassification of a form of grassland that covers large areas of the planet[25,26,37]. There are also increasing concerns over value-based definition error relating to the assumed socio-economic importance of grasslands, especially when they are classified as 'wasteland' despite the unique ecosystem services that grasslands provide[38,39]. Although we saw no evidence of this in our analysis, an extreme example is when Natural Climate Solution programmes target afforestation of areas with long legacies of grassland occurrence, justified by the often-incorrect assumption that forests sequester more C than grassland[40–42].

The unique value of our analysis is the creation of a high-accuracy collaborative fine-scale reference classification, which allowed us to assess product performance in ways typically not possible[11,43]. A lack of local data is the main reason why remotely sensed products are widely valued when attempting to quantify Sustainable Development Goals (SGDs[3]), but an absence of corroborating reference data remains a major threat to generalizability, policy performance and credibility[1,2,4,23,44]. Our work is one example of an attempt to bridge two disciplines undergoing rapid logistical and technical advancements that show great promise for assessment accuracy of SDGs: high-resolution remote sensing imagery for Earth-system modelling[6–10] and global-level collaborative ecosystem research targeting anthropogenic change[45–47]. Here we created a validation dataset relying on local knowledge that, in turn, served to calibrate high-resolution mapping products with a level of detail and spatial extent that no field researcher could ever mimic. Going forward, resolving mapping error will be increasingly important for quantifying grassland distribution and the contribution of grasslands towards environmental, social and economic sustainability[17,26,34,43]. There is an urgency to this work, as grasslands are one of the most transformed and at-risk biomes on the planet, with potentially large future shifts in occurrence from both losses to increased crop production, urban expansion, desertification and shrubification, and gains from deforestation and land retirement[20,48–50].

## Methods

### Analysed land cover products

We focused our analysis on the three LULC datasets with the highest resolution available at 10 m × 10 m pixels: World Cover (WC)[7], ESRI's Land Cover (LC)[6] and Dynamic World (DW) generated by GoogleDW[10]. Each dataset was accessed through the Google Earth Engine (GEE). As has been described[11], each product derives from Sentinel satellite imagery but utilizes it in different ways; for example, LC and DW share the same training dataset and are based on deep learning models, while WC is based on its own training data and random-forest modelling.

DW is updated in near-real-time with new sentinel imagery, while LC is intended to be updated annually, and WC was meant to be a standalone 2020 map but received an updated version in 2021. Given this latter issue, we directly compared the LULC products to each other for 2020–2021, the last year all three products were updated. Much of the data processing was, unless mentioned otherwise, scripted in a Python API via scripts running in Jupyter Notebooks using GEEmap software[51]. All metadata for this paper can be found in Supplementary Table 3. These served as the basis for all the results relating to error estimation for the three cover products. All coding used in the analyses is available via GitHub at https://github.com/B-vanzant/The-global-extent-of-the-grassland-biome-and-implications-for-the-terrestrial-carbon-sink.

## Researcher contact

We used a community-based effort to validate fine-scale 10-m resolution GEE pixels organized in primary sampling unit grids spanning the grassland biome (Fig. 1). Starting in June 2023, we made an open call for participation to validate grassland sites. All participants were grassland field ecologists or from related fields relating to land management, each of whom possessed local site-level knowledge of one or more grasslands such that they could validate the identity of different grassland and non-grassland cover types using remote sensing imagery. As shown in Supplementary Fig. 1, local knowledge is irreplaceable for validating image identity at high resolution, especially to distinguish among different types of grassland, or between grassland and non-grassland features such as recently harvested wheat fields, lawns or barren lands. This outreach resulted in a validation team of 157 experts who provided the coordinates of their own research sites, totalling to 504 unique grassland research locations spanning 60 countries across 6 continents including Greenland but excluding Antarctica (Supplementary Fig. 1). Collaborators were initially identified via the Nutrient Network (www.nutnet.org), DRAGnet (www.nutnet.org/dragnet) and DroughtNet (www.droughtnet.weebly.com), subsequently from mailing lists such as ECOLOG-L and OIKOS, using social media and word of mouth among researchers in the field of study. We also proactively contacted researchers in continents and geographic areas that were initially underrepresented (for example, eastern Europe, Africa, parts of Asia, South America and the Arctic).

The primary focus of the outreach was to create a geographically and bioclimatically representative distribution of locations across the grassland biome. As such, we did not emphasize grassland type (for example, tundra, savanna, open grassland) or any other type of attribute (for example, grazed/ungrazed, invaded by plants or uninvaded) when selecting our sites; the only requirement was that they be some form of grassland. Data submission was closed on 31 January 2024. As discussed below, our selection criteria for sites sought to balance estimation accuracy with the necessary practical constraints of finding local collaborators from all regions of the grassland biome[14]. Our inclusive acceptance criteria of any form of grassland—no matter the size, management history (for example, grazed/ungrazed, planted pasture), cover type (for example, invaded, uninvaded) or location—was one step of several that we utilized to minimize bias towards any specific cover type feature. The minimization of selection bias in these ways is an important requirement in both our assessment of classification accuracy by WC, LC and DW, and for our projected estimate of the most likely extent of the grassland biome[14]. It was also necessary because true random site selection for all of the Earth's terrestrial surface would disproportionately occur in non-grassland areas (because only ~22.8% of the Earth supports grassland) or select grassland in areas where local experts could not be identified.

## Selection of sampling units

Our evaluation centred on validating the LULC identity versus our 'reference classification' for 10 m × 10 m secondary sampling unit (SSU) pixels located within a 100 m × 100 m primary sampling unit (PSU) grid (Supplementary Fig. 2). We took multiple steps to ensure that all pixels

from our reference classification and the three LULC products were in full spatial agreement (that is, thus preventing geolocation error)[14]. We began by aligning the three LULC datasets (WC, LC and DW) through reprojection to EPSG:4326. Although in theory there could be shifting of pixel footprints, we assumed this not to be the case given that each LULC product derives from similar Sentinel imagery. For each of the 504 unique 'core' locations, the coordinates were added to a CSV file and then imported to GEE. Then, the PSU grids were overlaid on top of each provided grassland coordinate using the GEE covering grid tool, aligned directly with the pixels of the 10 m × 10 m LULC datasets.

Once we received the coordinates from the 504 core sites (that is, the main site where the researcher worked and/or was most familiar with), we randomly located an additional 10 sites within a 5-km radius (a 78-km² area) of each core location (with a few exceptions, see below). This distance was chosen after a preliminary trial run at the NutNet conference in the summer of 2023, where researchers decided that a 5-km radius provided the best spread of PSUs while still occurring in familiar geography. This was most critical in mountainous regions where large changes in vegetation type could occur over short distances with increasing altitude. In the end, this selection of additional grids resulted in 11 total PSU grids of 100 SSU pixels (1,100 pixels per site). The only times we did not generate the additional 10 grids per site were for locations where researchers provided large numbers of core sites within relatively small geographic regions; this involved five cases from three continents: Western Asia [Turkey (21 sites), Republic of Georgia and Armenia (21 sites)], North America [the state of Montana (31 western sites, 62 eastern sites)] and South America [Argentina (29 sites)]. These site clusterings closely resembled the PSU/SSU model that we used at all other sites, so we utilized these groupings in the same way.

The 10 additional grids had one selection criteria other than occurring with 5 km: they needed to have at least 1 of their 100 pixels ('1/100') overlapping with a pixel identified as 'grassland' by the LULC dataset with the highest global coverage (WC). This removed non-relevant grid placements (for example, grids placed entirely within other land classes such as bodies of water, cities or forests), while still allowing for placements where grids would have one or more grassland pixels. We selected WC for our '1/100' selection rule following a preliminary scan of grassland mapping by the three products, given that WC identified more than double their mapped estimates (~24.4%) as grassland (Supplementary Table 5) compared with LW and DW; the WC value was closest to the previously reported ranges for global grassland coverage (Table 1). In cases where some of the randomized PSU grids overlapped, the centroid coordinate that was last randomized was re-randomized (for example, if grids 4 and 7 were overlapping, 7 would be re-randomized) and relocated. The spatial constraints of this selection (within 5 km of the core site) meant that the site expert would probably be familiar with the vegetation characteristics of that location. We acknowledge that the relative proximity to the original coordinate could generate some degree of autocorrelation of the input values —neither our grassland nor grid selection protocols are truly randomized (see below). However, this approach allowed us to maximize the expert knowledge of local researchers, with the assumption that they would be sufficiently familiar with the local area to provide a qualified judgement for the additional randomized sites.

An important reason for adding these additional grids was to decrease bias by increasing the area sampled per location so that we could better test variability in mapping accuracy within and among sites. This was especially important given that some core sites were small and atypical compared to the surrounding landscape. The best example was when the core site was a conservation area or research management station embedded within a landscape dominated by crop fields, planted pasture, managed natural grassland, regenerating forest and/or settlement. The randomization of 10 PSUs surrounding the core site thus captured far more grassland and non-grassland cover types (27% of all pixels were non-grassland; Supplementary Table 3) within

regions of the Earth capable of supporting some type of grassland. This 'supporting some form of grassland' includes persistent anthropogenic grasslands within areas that might otherwise succeed to forest in the absence of human activity; such grasslands are common especially within many temperate regions of the world, and indeed 'pasture' constituted 9.1% (25,563/281,735) of all expert-identified grassland pixels (Supplementary Table 3).

## Image creation and sharing

We distributed images (png format) of the 11 PSU grids that were underlain with satellite imagery of very-high resolution (typically ~0.5–1 m; see 'Additional Methods' in Supplementary Information) to enable expert classification of grasslands (Supplementary Fig. 2). Imagery used was either taken from the Qgis Google satellite basemap or the Qgis Bing satellite basemap depending on the quality of imagery. The Bing basemap was used as a backup when the Google basemap was unusable based on one or more of the following criteria: (1) imagery was not sufficiently high resolution so that SSU pixels have discernible land cover, (2) ground cover was generally not visible (for example, heavy snow/cloud/shadow cover), or (3) artefacts and errors in basemap cover were present (for example, imagery warping or distortion in the basemap). In cases where researchers were unable to label the type of coverage for a given pixel with full confidence, they classified them either as 'U' for 'unfilled' or as '7' in cases where they knew the coverage was grassland but were unsure of the correct label (Supplementary Table 2), the latter applied to 4% of our SSUs. In cases where only a portion of the area within 5 km of the core site contained usable imagery, PSU grids were re-randomized until they fell within an area with suitable basemap coverage. This variability in basemap suitability is often an unappreciated and underestimated constraint when attempting to assemble global land cover basemaps using remote sensing; some regions of the Earth are comprehensively represented by multiple forms of remote images (for example, satellite- and airplane-derived) but others are not, especially in highly remote regions with low human population densities. In cases where both basemaps were unusable across the entire area of interest, these locations were removed from the study. This occurred rarely, with only 21 locations that signed up for the project being excluded due to unusable basemap data (leaving us with 504 locations).

Once the grids were finalized, images were exported from QGIS as PNG images at a scale of 1:1,900 for visualization while using the WGS84 pseudo-Mercator (EPSG: 4326) projection to aid in image visualization in higher latitudes. Finally, we sent each of our 157 researchers an overall regional map that showcased the relative locations of the core grid and the 10 additional grids within the 78 km² area. This helped participants to contextualize the location of the randomized sites relative to their provided research site to aid the annotation of the grids. The scale of these images depended on the distribution of the randomized grids within the area of interest and therefore was zoomed-in to the extent of all 11 grids rather than standardized at a specific scale. The provided 'core' research location was highlighted in the centre of each image, with each of the 10 additional randomized locations labelled with their assigned numeric value.

All site images and associated grids were hosted in a shared Google Drive where each researcher was given their own folder. There were 12 images per folder: the 11 grids plus the coarser-scale regional map. The breaking apart of images into 12 smaller-sized files was critical for co-principal investigators who lacked sufficient computation power or internet connection to download larger-sized image files.

## Site annotation and data cleaning

Along with the 11 PSU grid images and the 1 regional map image, a spreadsheet was provided to each researcher explaining the evaluation criteria (Supplementary Fig. 2 and Table 2). The spreadsheets contained highlighted data-entry grids, which corresponded in orientation to the 11 provided site grids. Also provided was a text-entry box for the researcher to describe their familiarity with the selected grid in case of extenuating circumstances that needed to be revisited (for example, uncertainty on pixel cover-type identity for whatever reason). One example might centre on uncertainty in the classification of lawns, which we subsequently classified as 'A = artificial'. Each of the grids was annotated by the corresponding researcher(s) through the criteria outlined in Supplementary Table 2. Spreadsheets were submitted through Dropbox in either a .xlsx or .csv file once completed by the researchers. The expert data were next cleaned using R, with errors standardized to our criteria on the basis of context clues (for example, a grid full of 1 s, accidentally having '12' inputted for a pixel, would be corrected to '1') and supporting comments.

## LULC area calculation

The area of each LULC dataset was calculated in GEE via the sum reducer, using a scale of 500 m due to computational limitations within the GEE platform. The 500-m pixel count for each dataset was converted to km² through simple division to give an approximate total area (for example, the coverage values for each platform given in Table 1).

## LULC data validation

Each labelled SSU-pixel score was extracted from the submitted spreadsheets into a standalone CSV file. It was then joined to the corresponding PSU grid raster files on the basis of the unique alphanumeric identifiers for each pixel (A1–J10; Supplementary Fig. 2) and its provided site name using QGIS Python scripting. The final CSV file of extracted expert annotations included 387,600 individually labelled SSU pixels, with 'U' (unfilled) cells only being attributed to fully unfilled PSUs in two cases. Each PSU grid was then used to extract the mapped values from the three LULC datasets for each unique corresponding SSU pixel. These data were then processed into three categories: (1) the cleaned data values from each dataset and the expert validation; (2) a binary dataset representing grassland and non-grassland pixels for each of the datasets; and (3) the expert 'reference classification' dataset and the LULC datasets simplified into one cohesive classification scheme following ref. 11 (see Supplementary Fig. 3 and Supplementary Table 6). The data were then directly compared (pixel identity in the reference classification versus pixel identity by each product) in R using the Kappa package. This allowed us to determine accuracy (user's and producer's accuracies for over- and underestimation) at three scales: PSU, location and overall. The PSU values represent the amount of agreement between the LULC datasets and the expertly labelled data for all pixels within each individual grid; the location scale values represent the agreement between the LULC and expert data across all 11 grids within a location; and the overall values represent the agreement between the LULC and expert data across all annotated grids (for example, Supplementary Table 3 and Fig. 2).

## Inferring corrected totals for global grassland area

'Confusion matrices' (that is, a data or 'error' matrix depicting accuracy based on the comparison of predicted versus our 'reference' classification) were next calculated both for overall grassland accuracy and for the 16 cover classes (see Supplementary Table 3). Apart from serving to demonstrate accuracy within and among LULC products, we sought to use these matrices to parameterize an adjusted estimate of actual grassland coverage following ref. 14 ('Good practices for estimating area and assessing accuracy of land change'), although with some modifications given that the true total area for grassland is unknown. Our calculations for total global grassland area thus centered on the accuracy of each LULC product (that is, differences in proportionality between the expert-validated pixels and the pixels classified as grassland in the LULC maps) and the area of global grassland that each depicts:

$$\text{Corrected global grassland area} = \text{LULC estimated area} \times \tag{1}$$

$$(\text{Validation proportion of pixels}/\text{Map proportion of pixels})$$

## Spatial distribution of site locations

A nearest-neighbour analysis was conducted to explore the degree of spatial clustering of our sites. We tested whether our 504 sites were clustered, dispersed or fully random on the basis of the average distances among sites, the distance to the nearest neighbouring sites and the expected mean distance if site locations within the grassland biome were fully random. Although one or more of our sites occur on most areas of the Earth supporting grassland, there are still some regions that appear to be undersampled or missed based on the WC grassland map (the yellow areas lacking dots in Fig. 1). Further, there was some clustering of researcher involvement despite our best efforts, especially towards North America (1,222 100 ×100 m grids) and Europe (968 grids) compared to 526 grids in Asia, 458 grids in South America, 361 grids in Africa and 209 grids in Oceania. Both factors could create potential bias, which could affect our estimated calculation of total grassland area after adjusting for over- and underestimation error. We based our calculations on the WC total for grasslands at 30.1 million km$^2$, given that we had determined its accuracy to be highest. Distances among research sites were calculated using the great circle tool from geopy (https://geopy.readthedocs.io/en/stable/), which calculates the shortest distance on the surface of an ellipsoidal model of the Earth. We found our sites to be dispersed, with a larger mean minimum distance of 171.51 km compared with an expected distance of 151.84 km ($Z$ = 4.62, s.e. = 4.26; $p$ < 0.0001; nearest-neighbour ratio 1.13). Thus, our sites show a tendency to be further apart than expected rather than the mean minimum distances expected of random or clustered distributions.

We also tested whether our sites were aligned with any underlying environmental parameters. We might expect this given that global grasslands are not randomly distributed but instead fall within a broad tension zone defined by areas too dry for forests but too wet for deserts, with the capacity to maintain disturbance regimes (for example, grazing, fire) that drive grassland persistence[20,32,50]. Further, environmental factors can shape site accessibility by researchers, which could create bias in site distribution. Finally, there may be pronounced differences in the sensitivity of the different LULC products to environmental factors that might help clarify their wide differences in performance (that is, Supplementary Table 3). We thus tested whether accuracy by cover-type product was associated with factors including climate, slope, elevation, latitude and remotely estimated maximum biomass (NDVI).

To do this, we conducted regression analyses using R v.4.4.1. We accounted for the nesting of the 11 grids within each site by using mixed effects models (MEMs) fit with the lmer function in the lme4 R library, with site included as a random effect. To account for correlation among the covariates (for example, climate, elevation, latitude), we used a multimodel approach to model selection using the dredge and model. avg functions in the MuMIn R library, because there could be multiple models with similar Akaike information criterion (AIC) values. We examined the suite of models within 4 AIC corrected (AICc) units of the top model (lowest AICc), and we standardized the input variables using the R arm library. For our data, we used:

- Average temperature (°C) over 30 years and average yearly total precipitation (mm) over 30 years extracted from WorldClim (http://worldclim.org)[52]
- Elevation as distance above sea level (m) extracted from ALOS DSM: Global 30 m v.3.2
- Site aspect as the direction of hillslope (degrees: 0–360), while slope as the overall hillslope of land area (degrees: 0–90); both derived from ALOS DSM: Global 30 m v.3.2.(https://developers.google.com/earth-engine/datasets/catalog/JAXA_ALOS_AW3D30_V3_2)
- Normalized difference vegetation index (NDVI) extracted from MOD13A1.061 Terra Vegetation Indices 16-Day Global 500 m (https://developers.google.com/earth-engine/datasets/catalog/MODIS_061_MOD13A1)

Finally, we visually depicted our distribution of grids relative to temperature and precipitation data by creating Whittaker biomes plots using the PlotBiomes package (Supplementary Fig. 5). The goal was to assess whether certain biomes might be underrepresented, which we found not to be the case on the basis of visual assessment.

## Reporting summary

Further information on research design is available in the Nature Portfolio Reporting Summary linked to this article.

## Data availability

All data and materials used in the analysis are available from the corresponding authors on request.

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

## Acknowledgements

We thank each of the researchers who have contributed data and ideas to this paper. This study was largely funded by the Canada First Research Excellence Fund—University of Guelph ('Food from Thought'), with support from the Natural Sciences and Engineering Research Council of Canada (A.S.M.). M.B.S. acknowledges funding from the Swedish Research Council (2021-05767), FORMAS (2020-01073) and the European Union's Horizon Program project ILLUQ (no. 101133587). Funding was also provided to E.W.S. and E.T.B. by the National Science Foundation Research Coordination Network (NSF-DEB-1042132) and the Long-Term Ecological Research (NSF-DEB-1234162 to Cedar Creek LTER) programmes, and the Institute on the Environment (DG-0001-13). Y.M.B. acknowledges financial support from Research Ireland, Northern Ireland's Department of Agriculture, Environment and Rural Affairs (DAERA), UK Research and Innovation (UKRI) via the International Science Partnerships Fund (ISPF) under grant number [22/CC/11103] at the Co-Centre for Climate + Biodiversity + Water. N.E. was supported by the German Centre for Integrative Biodiversity Research (iDiv), Halle-Jena-Leipzig iDiv funded by the German Research Foundation (DFG– FZT 118, 202548816), and funding by the DFG (Ei 862/29-1). S.C.P. acknowledges funding from NSF OCE-1832178.

## Author contributions

A.S.M., M.B.S., J.S. and B.V., with D.N., S. Bagchi and T.O.M., conceptualized the project. B.V., with A.S.M. and M.B.S., designed the methodology. B.V., with A.S.M., M.B.S. and J.S., conducted the investigations. All authors gathered the data. B.V., with A.S.M., M.B.S. and J.S., performed the visualization. A.S.M., with M.B.S., acquired the funding. A.S.M., with M.B.S., J.S., S. Bagchi, D.N. and T.O.M., administered the project. A.S.M., with M.B.S. and J.S., supervised the project. A.S.M., with B.V., M.B.S., J.S., E.W.S. and E.T.B., wrote the original draft of the paper. All authors reviewed and edited the paper.

## Funding

## Competing interests

The authors declare no competing interests.

## Additional information

**Correspondence and requests for materials** should be addressed to A. S. MacDougall or M. B. Siewert.

A. S. MacDougall ®[1]✉, B. Vanzant[1], J. Sulik ®[2], S. Bagchi[3], D. Naidu ®[3,4], T. O. Muraina ®[5,6], E. W. Seabloom ®[7], E. T. Borer ®[7], P. Wilfahrt[7], I. Slette[7], J. L. Hierro ®[8], D. E. Pearson[9], M. Abedi ®[10], M. Akasaka ®[11], J. Alberti ®[12], A. Aleksanyan ®[13], A. A. Amisu[14], T. M. Anderson[15], C. A. Arnillas[16], M. Ayer ®[17], J. D. Bakker ®[18], S. Basant[19], S. Basto ®[20], L. Biederman ®[21], K. J. Bloodworth[22], F. Boscutti ®[23,24], E. H. Boughton ®[25], C. M. Bruschetti[12], H. L. Buckley ®[26], Y. M. Buckley ®[27], M. N. Bugalho ®[28], M. C. Caldeira ®[29], G. Campetella ®[30], N. Cannone ®[31,32], M. Carbognani ®[33], C. Carbutt ®[34,35], M. A. Carniello[36], M. Cervellini[37], T. Chaudhary ®[38], Q. Chen ®[39], A. T. Clark[40], S. Cousins[41], M. Dalle Fratte[42], N. J. Day[43], B. Deák ®[44], J. Dietrich[45], A. Dixon[46,47], N. Eisenhauer ®[48,49], K. J. Elgersma ®[50], O. Eren[51], A. Eskelinen ®[52,53], C. Estrada ®[54], P. A. Fay[55], G. Fayvush[13], K. C. Flynn[55], D. García Meza[56], D. Gargano[57], L. Gherardi ®[58], N. T. Girkin[59], L. González[60], P. Graff ®[61], L. W. C. Hagenberg[45], A. H. Halbritter ®[62], N. A. Havrilchak ®[63], N. Herdoiza[64], E. Hersch-Green ®[65], K. Hopping ®[66], A. Jentsch ®[67], S. O. Jimoh[68], J. Kerby[69,70], K. Kirkman ®[71], J. M. H. Knops[72], S. E. Koerner ®[73], A. Koltz[74], K. J. Komatsu ®[73], B. I. Koura[75], S. Kruse ®[76], L. Laanisto ®[77], L. S. Lannes ®[78], W. Li ®[79], M. Liang[80], A. Lkhagva[81], L. López-Olmedo ®[82], P. Lorenzo ®[83], C. J. Lortie[84], A. Loydi[85], W. Luo[86], P. Macek ®[77,87], F. Malfasi[32,88], P. Mariotte ®[89], J. P. Martina ®[6], A. Martínez-Blancas ®[90,91], H. Martinson ®[92], C. Martorell ®[82],

Article

J. A. Meave [82], S. Medina-Villar[93,94], K. Z. Mganga [64], J. Monsimet [45], A. N. Nerlekar[90], S. Niu [95], T. Ohlert[96,97], I. Oliveras Menor [98,99], G. R. Oñatibia [100], Y. K. Ortega [9], B. Osborne[101], S. Palpurina [102], J. Pascual[12], S. C. Pennings[103], E. Pérez-García [82], P. L. Peri [104], M. Petit Bon [105,106], A. Petraglia [33], F. Pijcke [45], S. M. Prober[107], R. E. Quiroga [108], J. I. Ramirez [45], S. Reed[109], B. H. P. Rosado[110], C. Roscher [49,111], D. W. Rowley [55], I. Sereda[112], D. M. Small[113], N. G. Smith[114], Y. Song [115], C. Stevens[116], L. E. Suarez Jimenez[117], M. te Beest [64,118,119], M. Tedder[120], R. S. Terry[73], K. S. Thornton[121], D. Tian [95], G. Titcomb[122], O. Valkó [44], G. F. 'Ciska' Veen [123], R. Virtanen [52], E. A. R. Welti[124], G. R. Wheeler[125], A. A. Wolf [126], P. Wolff[67], A. L. Young [73], H. S. Young[127], L. H. Zeglin[128], K. Zhu[115], S. Zong[129] & M. B. Siewert [45,130] ✉

[1]Department of Integrative Biology, University of Guelph, Guelph, Ontario, Canada. [2]Department of Plant Agriculture, University of Guelph, Guelph, Ontario, Canada. [3]Centre for Ecological Sciences, Indian Institute of Science, Bengaluru, India. [4]School of Environment and Sustainability, Indian Institute for Human Settlements, Bengaluru, India. [5]Department of Animal Health and Production, Oyo State College of Agriculture and Technology, Oyo State, Nigeria. [6]Department of Biology, Texas State University, San Marcos, TX, USA. [7]Department of Ecology, Evolution, and Behavior, University of Minnesota, St. Paul, MN, USA. [8]Facultad de Ciencias Exactas y Naturales, Universidad Nacional de La Pampa (UNLPam) and Laboratorio de Ecología, Biogeografía y Evolución Vegetal (LEByEV), Instituto de Ciencias de la Tierra y Ambientales de La Pampa (INCITAP), Consejo Nacional de Investigaciones Científicas y Técnicas (CONICET)-UNLPam, Santa Rosa, La Pampa, Argentina. [9]Rocky Mountain Research Station, United States Forest Service, Missoula, MT, USA. [10]Department of Range Management, Faculty of Natural Resources and Marine Sciences, Tarbiat Modares University, Noor, Iran. [11]Tokyo University of Agriculture and Technology, Tokyo, Japan. [12]Instituto de Investigaciones Marinas y Costeras (IIMyC), FCEyN-UNMDP – CONICET, Mar del Plata, Argentina. [13]Institute of Botany aft. A.Takhtajyan NAS, Yerevan, Republic of Armenia. [14]Federal University of Agriculture, Abeokuta, Nigeria. [15]Wake Forest University, Winston-Salem, NC, USA. [16]University of Toronto–Scarborough, Scarborough, Ontario, Canada. [17]Department of National Parks and Wildlife Conservation, Kathmandu, Nepal. [18]School of Environmental and Forest Sciences, University of Washington, Seattle, WA, USA. [19]Department of Natural Resources and Environmental Science, University of Nevada, Reno, NV, USA. [20]Unidad de Ecología y Sistemática, Departamento de Biología, Facultad de Ciencias, Pontificia Universidad Javeriana, Bogotá, Colombia. [21]Department of Ecology, Evolution and Organismal Biology, Iowa State University, Ames, IA, USA. [22]University of Maryland, College Park, MD, USA. [23]Department of Agricultural, Food, Environmental and Animal Sciences, University of Udine, Udine, Italy. [24]National Biodiversity Future Center (NBFC), Palermo, Italy. [25]Archbold Biological Station, Venus, FL, USA. [26]School of Science, Auckland University of Technology, Auckland, New Zealand. [27]Co-Centre for Climate, Biodiversity, and Water, School of Natural Sciences, Trinity College Dublin, Dublin 2, Ireland. [28]Center for Applied Ecology "Prof. Baeta Neves" (CEABN-InBIO), School of Agriculture, University of Lisbon, Lisbon, Portugal. [29]Forest Research Centre, Associate Laboratory TERRA, School of Agriculture, University of Lisbon, Lisbon, Portugal. [30]Torricchio Nature Reserve, University of Camerino, Torina, Italy. [31]Department of Theoretical and Applied Sciences, University of Insubria, Varese, Italy. [32]Climate Change Research Center, University of Insubria, Como, Italy. [33]Department of Chemistry, Life Sciences and Environmental Sustainability, Parma, Italy. [34]School of Life Sciences, University of KwaZulu-Natal, Scottsville, South Africa. [35]Scientific Services, Ezemvelo KZN Wildlife, Cascades, South Africa. [36]Institute de Recherche pour le Developpement, Marseille, France. [37]Torricchio Nature Reserve, University of Camerino, Camerino, Italy. [38]Texas State University, San Marcos, TX, USA. [39]German Centre for Integrative Biodiversity Research (iDiv), Leipzig, Germany. [40]Department of Biology, University of Graz, Graz, Austria. [41]Department of Physical Geography, Stockholm University, Stockholm, Sweden. [42]Department of Biotechnology and Life Science, University of Insubria, Varese, Italy. [43]School of Biological Sciences, Victoria University of Wellington, Wellington, New Zealand. [44]Lendület Seed Ecology Research Group, Institute of Ecology and Botany, HUN-REN Centre for Ecological Research, Vácrátót, Hungary. [45]Department of Ecology and Environmental Science (EMG), Umeå University, Umeå, Sweden. [46]World Wildlife Fund, Washington, DC, USA. [47]Great Plains Program, Lawton, USA. [48]German Centre for Integrative Biodiversity Research (iDiv) Halle-Jena-Leipzig, Leipzig, Germany. [49]Institute of Biology, Leipzig University, Leipzig, Germany. [50]University of Northern Iowa, Cedar Falls, IA, USA. [51]Biyoloji Bölümü, Fen Fakültesi, Aydın Adnan Menderes Üniversitesi, Aydın, Turkey. [52]Ecology and Genetics Unit, University of Oulu, Oulu, Finland. [53]German Centre for Integrative Biodiversity Research iDiv, Leipzig, Germany. [54]Department of Life Sciences, Imperial College London, London, UK. [55]Soil and Water Research Laboratory, Agricultural Research Service, Grassland, USDA, Temple, TX, USA. [56]Depto. de Ecología y Recursos Naturales, Facultad de Ciencias, Universidad Nacional Autónoma de México, Mexico City, Mexico. [57]Department of Biology, Ecology and Earth Sciences, University of Calabria, Arcavacata, Italy. [58]Department of Environmental Sciences, Policy, and Management, University of California, Berkeley, USA. [59]School of Biosciences, University of Nottingham, Nottingham, UK. [60]CISPAC, Departamento de Bioloxía Vexetal e Ciencia do Solo, Facultade de Bioloxía, Universidade de Vigo, Vigo, Spain. [61]CONICET and Agencia de Extensión Rural Coronel Suárez, EEA Cesareo Naredo, Instituto Nacional de Tecnología Agropecuaria (INTA), Argentina; Facultad de Agronomía, Universidad de Buenos Aires, Buenos Aires, Argentina. [62]Department of Biological Sciences and Bjerknes Center for Climate Research, University of Bergen, Bergen, Norway. [63]USDA Agricultural Research Service, Reno, NV, USA. [64]Copernicus Institute of Sustainable Development, Utrecht University, Utrecht, the Netherlands. [65]Michigan Technological University, Houghton, MI, USA. [66]Human-Environment Systems, Boise State University, Boise, ID, USA. [67]Disturbance Ecology and Vegetation Dynamics, University of Bayreuth, Bayreuth, Germany. [68]Department of Botany, University of Wyoming, Laramie, WY, USA. [69]Scott Polar Research Institute, Department of Geography, University of Cambridge, Cambridge, UK. [70]Institute of Arctic Studies, Dartmouth College, Hanover, NH, USA. [71]Grassland Science, University of KwaZulu-Natal, Pietermaritzburg, South Africa. [72]Health and Environmental Sciences, Xi'an Jiaotong-Liverpool University, Suzhou, Jiangsu Province, China. [73]University of North Carolina Greensboro, Greensboro, USA. [74]University of Texas at Austin, Austin, USA. [75]Ecole de Gestion et d'Exploitation des Systèmes d'Elevage, Université Nationale d'Agriculture, Kétou, Benin. [76]Alfred Wegener Institute, Helmholtz Centre for Polar and Marine Research, Bremerhaven, Germany. [77]Chair of Biodiversity and Nature Tourism; Estonian University of Life Sciences, Tartu, Estonia. [78]São Paulo State University (UNESP), School of Engineering, Department of Biology and Animal Science, Ilha Solteira, São Paulo, Brazil. [79]Ministry of Education Key Laboratory of Ecology and Resource Use of the Mongolian Plateau, and Inner Mongolia Key Laboratory of Grassland Ecology, and Observation and Research Station for the Typical Steppe Ecosystem of the Ministry of Education, School of Ecology and Environment, Inner Mongolia University, Hohhot, China. [80]Cedar Creek Ecosystem Science Reserve, University of Minnesota, Minneapolis, USA. [81]Department of Biology, School of Arts and Sciences, National University of Mongolia, Ulan Bator, Mongolia. [82]Depto. de Ecología y Recursos Naturales Circuito Exterior s/n, Facultad de Ciencias, Universidad Nacional Autónoma de México, Cd. Universitaria, Mexico City, Mexico. [83]Centre for Functional Ecology (CFE)—Science for People and the Planet, Associate Laboratory TERRA, Department of Life Sciences, University of Coimbra, Coimbra, Portugal. [84]York University, Toronto, Canada.

[85]CERZOS CONICET-UNS and DBByF UNS, Buenos Aires, Argentina. [86]Erguna Forest-Steppe Ecotone Research Station, Institute of Applied Ecology, Chinese Academy of Sciences, Beijing, China. [87]Institute of Hydrobiology, Biology Centre of Czech Academy of Sciences, Branišovská, Czech Republic. [88]Department of Science and High Technology, University of Insubria, Como, Italy. [89]Agroscope, Grazing Systems, Posieux, Switzerland. [90]Department of Plant Biology, Michigan State University, East Lansing, MI, USA. [91]Ecology, Evolution, and Behavior Program, Michigan State University, East Lansing, MI, USA. [92]McDaniel College, Westminster, MD, USA. [93]Department of Plant Protection, National Center for Agricultural and Food Research and Technology (INIA-CSIC), Madrid, Spain. [94]Department of Natural Sciences, Saint Luis University - Madrid Campus, Madrid, Spain. [95]Institute of Geographic Sciences and Natural Resources Research, Chinese Academy of Sciences, Beijing, China. [96]Department of Biology, Colorado State University, Fort Collins, CO, USA. [97]Department of Biology, University of New Mexico, Albuquerque, NM, USA. [98]AMAP, University of Montpellier, CIRAD, IRD, CNRS, INRAE, Montpellier, France. [99]School of Geography and the Environment, University of Oxford, Oxford, UK. [100]Instituto de Investigaciones Fisiológicas y Ecológicas Vinculadas a la Agricultura (IFEVA), Facultad de Agronomía, Universidad de Buenos Aires and CONICET, Buenos Aires, Argentina. [101]Utah State University, Logan, UT, USA. [102]National Museum of Natural History, Bulgarian Academy of Sciences, Sofia, Bulgaria. [103]Department of Biology and Biochemistry, University of Houston, Houston, TX, USA. [104]InstitutoNacional de Tecnología Agropecuaria (INTA), Southern Patagonia National University (UNPA)-CONICET, Santa Cruz, Argentina. [105]Department of Wildland Resources, Quinney College of Natural Resources and Ecology Center, Utah State University, Logan, UT, USA. [106]Department of Applied Ecology, College of Agriculture and Life Sciences, North Carolina State University, Raleigh, NC, USA. [107]CSIRO Environment, Canberra, Australian Capital Territory, Australia. [108]Instituto Nacional de Tecnología Agropecuaria, Buenos Aires, Argentina. [109]Southwest Biological Science Center, United States Geological Survey, Moab, UT, USA. [110]Department of Ecology, State University of Rio de Janeiro (UERJ), Rio de Janeiro, Rio de Janeiro, Brazil. [111]UFZ, Helmholtz Centre for Environmental Research, Physiological Diversity, Leipzig, Germany. [112]Center for Ecological-Noosphere Studies NAS Republic of Armenia, Yerevan, Armenia. [113]Washington College, Chestertown, MD, USA. [114]Department of Biological Sciences, Texas Tech University, Lubbock, TX, USA. [115]School for Environment and Sustainability, University of Michigan, Ann Arbor, MI, USA. [116]Lancaster Environment Centre, Lancaster University, Lancaster, UK. [117]Universidad Internacional del Trópico Americano (Unitrópico), Casanare, Colombia. [118]Centre for African Conservation Ecology, Nelson Mandela University, Gqeberha, South Africa. [119]South African Environmental Observation Network, Grasslands-Forests-Wetlands Node, Pietermaritzburg, South Africa. [120]School of Agriculture and Science, University of KwaZulu-Natal, Pietermaritzburg, South Africa. [121]Washington College Center for Environment and Society, Chestertown, MD, USA. [122]Colorado State University, Fort Collins, CO, USA. [123]Netherlands Institute of Ecology, Wageningen, the Netherlands. [124]Great Plains Science Program, Smithsonian Institution, Bozeman, MT, USA. [125]Michigan Technological University; University of Nebraska-Lincoln, Lincoln, USA. [126]University of Texas, Austin, TX, USA. [127]University of California, Santa Barbara, CA, USA. [128]Kansas State University, Manhattan, KS, USA. [129]Key Laboratory of Geographical Processes and Ecological Security in Changbai Mountains, Ministry of Education, School of Geographical Sciences, Northeast Normal University, Changchun, China. [130]Climate Impacts Research Centre (CIRC), Umeå University, Umeå, Sweden. ✉e-mail: asm@uoguelph.ca; matthias.siewert@umu.se

# Reporting Summary

Please do not complete any field with "not applicable" or n/a.  Refer to the help text for what text to use if an item is not relevant to your study.
For final submission: please carefully check your responses for accuracy; you will not be able to make changes later.

## Statistics

For all statistical analyses, confirm that the following items are present in the figure legend, table legend, main text, or Methods section.

| n/a | Confirmed | |
|---|---|---|
| ☐ | [x] | The exact sample size (*n*) for each experimental group/condition, given as a discrete number and unit of measurement |
| ☐ | [x] | A statement on whether measurements were taken from distinct samples or whether the same sample was measured repeatedly |
| ☐ | [x] | The statistical test(s) used AND whether they are one- or two-sided<br>*Only common tests should be described solely by name; describe more complex techniques in the Methods section.* |
| ☐ | [x] | A description of all covariates tested |
| ☐ | [x] | A description of any assumptions or corrections, such as tests of normality and adjustment for multiple comparisons |
| ☐ | [x] | A full description of the statistical parameters including central tendency (e.g. means) or other basic estimates (e.g. regression coefficient) AND variation (e.g. standard deviation) or associated estimates of uncertainty (e.g. confidence intervals) |
| ☐ | [x] | For null hypothesis testing, the test statistic (e.g. *F*, *t*, *r*) with confidence intervals, effect sizes, degrees of freedom and *P* value noted<br>*Give P values as exact values whenever suitable.* |
| [x] | ☐ | For Bayesian analysis, information on the choice of priors and Markov chain Monte Carlo settings |
| ☐ | [x] | For hierarchical and complex designs, identification of the appropriate level for tests and full reporting of outcomes |
| [x] | ☐ | Estimates of effect sizes (e.g. Cohen's *d*, Pearson's *r*), indicating how they were calculated |

*Our web collection on statistics for biologists contains articles on many of the points above.*

## Software and code

Policy information about availability of computer code

| Data collection | described in Methods |
|---|---|
| Data analysis | described in Methods |

For manuscripts utilizing custom algorithms or software that are central to the research but not yet described in published literature, software must be made available to editors and reviewers. We strongly encourage code deposition in a community repository (e.g. GitHub). See the Nature Portfolio guidelines for submitting code & software for further information.

## Data

Policy information about availability of data

All manuscripts must include a data availability statement. This statement should provide the following information, where applicable:

- Accession codes, unique identifiers, or web links for publicly available datasets
- A description of any restrictions on data availability
- For clinical datasets or third party data, please ensure that the statement adheres to our policy

yes; done

## Research involving human participants, their data, or biological material

Policy information about studies with human participants or human data. See also policy information about sex, gender (identity/presentation), and sexual orientation and race, ethnicity and racism.

| | |
|---|---|
| Reporting on sex and gender | NA |
| Reporting on race, ethnicity, or other socially relevant groupings | |
| Population characteristics | |
| Recruitment | |
| Ethics oversight | |

Note that full information on the approval of the study protocol must also be provided in the manuscript.

# Field-specific reporting

Please select the one below that is the best fit for your research. If you are not sure, read the appropriate sections before making your selection.

☐ Life sciences ☐ Behavioural & social sciences [X] Ecological, evolutionary & environmental sciences

For a reference copy of the document with all sections, see nature.com/documents/nr-reporting-summary-flat.pdf

# Life sciences study design

All studies must disclose on these points even when the disclosure is negative.

| | |
|---|---|
| Sample size | |
| Data exclusions | |
| Replication | |
| Randomization | |
| Blinding | |

# Behavioural & social sciences study design

All studies must disclose on these points even when the disclosure is negative.

| | |
|---|---|
| Study description | |
| Research sample | |
| Sampling strategy | |
| Data collection | |
| Timing | |
| Data exclusions | |
| Non-participation | |
| Randomization | |

# Ecological, evolutionary & environmental sciences study design

All studies must disclose on these points even when the disclosure is negative.

| | |
|---|---|
| Study description | DONE |
| Research sample | DONE |
| Sampling strategy | DONE |
| Data collection | DONE |
| Timing and spatial scale | DONE |
| Data exclusions | DONE |
| Reproducibility | DONE |
| Randomization | DONE |
| Blinding | NA |

Did the study involve field work? ☐ Yes ☒ No

## Field work, collection and transport

| | |
|---|---|
| Field conditions | |
| Location | |
| Access & import/export | |
| Disturbance | |

# Reporting for specific materials, systems and methods

We require information from authors about some types of materials, experimental systems and methods used in many studies. Here, indicate whether each material, system or method listed is relevant to your study. If you are not sure if a list item applies to your research, read the appropriate section before selecting a response.

## Materials & experimental systems

| n/a | Involved in the study |
|---|---|
| ☒ | ☐ Antibodies |
| ☒ | ☐ Eukaryotic cell lines |
| ☒ | ☐ Palaeontology and archaeology |
| ☒ | ☐ Animals and other organisms |
| ☒ | ☐ Clinical data |
| ☒ | ☐ Dual use research of concern |
| ☒ | ☐ Plants |

## Methods

| n/a | Involved in the study |
|---|---|
| ☒ | ☐ ChIP-seq |
| ☒ | ☐ Flow cytometry |
| ☒ | ☐ MRI-based neuroimaging |

## Antibodies

| | |
|---|---|
| Antibodies used | NA |
| Validation | |

# Eukaryotic cell lines

Policy information about cell lines and Sex and Gender in Research

| | |
|---|---|
| Cell line source(s) | NA |
| Authentication | |
| Mycoplasma contamination | |
| Commonly misidentified lines (See ICLAC register) | |

# Palaeontology and Archaeology

| | |
|---|---|
| Specimen provenance | NA |
| Specimen deposition | |
| Dating methods | |

☐ Tick this box to confirm that the raw and calibrated dates are available in the paper or in Supplementary Information.

| | |
|---|---|
| Ethics oversight | |

Note that full information on the approval of the study protocol must also be provided in the manuscript.

# Animals and other research organisms

Policy information about studies involving animals; ARRIVE guidelines recommended for reporting animal research, and Sex and Gender in Research

| | |
|---|---|
| Laboratory animals | NA |
| Wild animals | |
| Reporting on sex | |
| Field-collected samples | |
| Ethics oversight | |

Note that full information on the approval of the study protocol must also be provided in the manuscript.

# Clinical data

Policy information about clinical studies

All manuscripts should comply with the ICMJE guidelines for publication of clinical research and a completed CONSORT checklist must be included with all submissions.

| | |
|---|---|
| Clinical trial registration | NA |
| Study protocol | |
| Data collection | |
| Outcomes | |

# Dual use research of concern

Policy information about dual use research of concern

## Hazards

Could the accidental, deliberate or reckless misuse of agents or technologies generated in the work, or the application of information presented in the manuscript, pose a threat to:

| No | Yes | |
|----|-----|--|
| X | ☐ | Public health |
| X | ☐ | National security |
| X | ☐ | Crops and/or livestock |
| X | ☐ | Ecosystems |
| X | ☐ | Any other significant area |

## Experiments of concern

Does the work involve any of these experiments of concern:

| No | Yes | |
|----|-----|--|
| X | ☐ | Demonstrate how to render a vaccine ineffective |
| X | ☐ | Confer resistance to therapeutically useful antibiotics or antiviral agents |
| X | ☐ | Enhance the virulence of a pathogen or render a nonpathogen virulent |
| X | ☐ | Increase transmissibility of a pathogen |
| X | ☐ | Alter the host range of a pathogen |
| X | ☐ | Enable evasion of diagnostic/detection modalities |
| X | ☐ | Enable the weaponization of a biological agent or toxin |
| X | ☐ | Any other potentially harmful combination of experiments and agents |

# Plants

| Seed stocks | NA |
|----|----|
| Novel plant genotypes | |
| Authentication | |

# ChIP-seq

## Data deposition

☐ Confirm that both raw and final processed data have been deposited in a public database such as GEO.

☐ Confirm that you have deposited or provided access to graph files (e.g. BED files) for the called peaks.

| Data access links *May remain private before publication.* | |
|----|----|
| Files in database submission | |
| Genome browser session (e.g. UCSC) | |

## Methodology

| Replicates | |
|----|----|
| Sequencing depth | |
| Antibodies | |
| Peak calling parameters | |
| Data quality | |

| Software | |
|---|---|

# Flow Cytometry

## Plots

Confirm that:

☐ The axis labels state the marker and fluorochrome used (e.g. CD4-FITC).

☐ The axis scales are clearly visible. Include numbers along axes only for bottom left plot of group (a 'group' is an analysis of identical markers).

☐ All plots are contour plots with outliers or pseudocolor plots.

☐ A numerical value for number of cells or percentage (with statistics) is provided.

## Methodology

| Sample preparation | |
|---|---|
| Instrument | |
| Software | |
| Cell population abundance | |
| Gating strategy | |

☐ Tick this box to confirm that a figure exemplifying the gating strategy is provided in the Supplementary Information.

# Magnetic resonance imaging

## Experimental design

| Design type | |
|---|---|
| Design specifications | |
| Behavioral performance measures | |

| Imaging type(s) | |
|---|---|
| Field strength | |
| Sequence & imaging parameters | |
| Area of acquisition | |

Diffusion MRI  ☐ Used  ☐ Not used

## Preprocessing

| Preprocessing software | |
|---|---|
| Normalization | |
| Normalization template | |
| Noise and artifact removal | |
| Volume censoring | |

## Statistical modeling & inference

| Model type and settings | |
|---|---|
| Effect(s) tested | |

Specify type of analysis: ☐ Whole brain ☐ ROI-based ☐ Both

Statistic type for inference

(See Eklund et al. 2016)

Correction

## Models & analysis

| n/a | Involved in the study |
|---|---|
| ☒ ☐ | Functional and/or effective connectivity |
| ☒ ☐ | Graph analysis |
| ☒ ☐ | Multivariate modeling or predictive analysis |

Functional and/or effective connectivity

Graph analysis

Multivariate modeling and predictive analysis

