## [Peer Review File · Nature Ecology & Evolution]

The global extent of the grassland biome and implications for the terrestrial carbon sink

Corresponding Author: Dr Andrew MacDougall

Version 0:

Decision Letter:

9th April 2025

Dear Dr MacDougall,

Your manuscript entitled "Global extent of the grassland biome at high spatial resolution" has now been seen by two reviewers, whose comments are provided below. As you will see, the reviewers find your work of potential interest but raise a number of concerns to address. We will therefore need to see your responses to the criticisms raised, along with a revised manuscript, before we can reach a final decision regarding publication.

* Please highlight all changes in the manuscript text file and provide it in Microsoft Word format, with line numbers.

* If you have not done so already please begin to revise your manuscript so that it conforms to our Article format instructions at <http://www.nature.com/natecolvol/info/final-submission>. Refer also to any guidelines provided in this letter.

* Extended Data Figures - please ensure that any supplementary figures and tables that are crucial to the manuscript's conclusions are converted into Extended Data figures and tables to increase visibility of these data. Extended Data figures and tables are online-only (present in the online PDF and full-text HTML versions of the paper), peer-reviewed display items that provide essential background to the article but are not included in the main article due to space constraints. A maximum of ten Extended Data display items (figures and tables) is permitted.

Link Redacted

We hope to receive your revised manuscript within two months. If you cannot send it within this time, please let us know. We will be happy to consider your revision so long as nothing similar has been accepted for publication at Nature Ecology & Evolution or published elsewhere.

Nature Ecology & Evolution is committed to improving transparency in authorship. As part of our efforts in this direction, we

are now requesting that all authors identified as 'corresponding author' on published papers create and link their Open Researcher and Contributor Identifier (ORCID) with their account on the Manuscript Tracking System (MTS), prior to acceptance. ORCID helps the scientific community achieve unambiguous attribution of all scholarly contributions. You can create and link your ORCID from the home page of the MTS by clicking on 'Modify my Springer Nature account'. For more information please visit www.springernature.com/orcid.

Please do not hesitate to contact me if you have any questions or would like to discuss these revisions further. We look forward to seeing the revised manuscript.

[redacted]

Reviewers' comments:

Reviewer #1 (Remarks to the Author):

While the implications of using different maps on estimates of carbon stocks warrant investigation, the manuscript suffers from shortcomings that need to be addressed before publication can be considered.

First, the manuscript is not aligned with the remote sensing literature that have discussed issues of accuracy, bias, and uncertainty for decades. The notion that classification "errors ... must be rectified" if remote sensing-based products are to be used for decision-making, for reporting areas, in subsequent analysis, etc., has been emphasized by the remote sensing community for decades. Multiple articles on scientific inference versus pretty satellite images, issues of sampling techniques, and protocols for unbiased estimation of areas, have been published in the remote sensing literature; the manuscript in question is completely disconnected from this wealth of literature. (Much of this literature has been published in the journal *Remote Sensing of Environment* and it is telling that the manuscript in question does not cite a single article published in that journal.) The authors state that their "work differs [because] images are validated by experts with direct on-the-ground familiarity of their sites" – again, it is strongly recommended by the RS community that not only have land cover experts collect reference observations but to estimate reference data uncertainty and interpreter variability.

Second, if the authors aim to provide a global unbiased sampling-based estimate of the area of grassland (and estimates of commission and omission error), the reference sample data must be collected under a probability design with attention to the inclusion probabilities if using traditional design-based estimators (with the estimators corresponding to the sampling design). If the sample data were not collected under a probability design, it is necessary to invoke a model-based inference framework. But the manuscript provides no discussion of sampling designs, construction of estimators, or of inference frameworks. While there's a long description of "Generation of ground truth grids," it's unclear if this approach conforms to a probability design.

Reviewer #2 (Remarks to the Author):

This is a very interesting paper that addresses a common issue affecting global land cover maps - how accurate are they, and how can the most accurate one be determined, and as the authors highlight, in the absence of being able to answer those questions there is a tendency to quote what someone else has written because that has already been accepted by peer review as being correct!

The paper is well written, and synthesises a large amount of information succinctly - clearly a lot of time and effort was invested in trying to gather the individual experts and generate the image files for them to ensure global coverage in so far as possible. Most of the comments below are for clarification and discussion rather than expecting further work to be conducted.

Turning first to the main paper.

The abstract lacks detail on the results of comparing the land cover maps, and focuses instead on the carbon storage, which is not the focus of the paper or its title. The abstract would benefit from a total rewrite with an emphasis on the discrepancies and challenges of the existing grassland maps, and the new estimate of grassland cover - especially if the authors want to be quoted as the new Loveland et al (2000) authoritative estimate of grassland extent.

The main text needs some additional clarification and discussion in places as noted below.

Around line 240 - although the three products are discussed in detail in the Supplementary Materials section, it would be helpful to have a few extra comments on the nature of the products given that they form the basis of the paper, e.g. years of availability, frequency of update, method of generation etc.

Line 252, why are the challenges particularly pronounced for grasslands?

Line 258, close the brackets

Line 272, MODIS is a sensor on the TERRA and AQUA satellites

Line 339, one other cause of the discrepancy is the spatial resolution of the datasets cited in the 2000 papers e.g. Loveland et al use AVHRR 1km data which is invariably going to result in aggregation of smaller units - modifiable areal unit problem scale effect

Line 373, 379 and elsewhere, there is inconsistency in the use of images, areas, cells and pixels - I recommend removing all references to the individual cells as images or areas as that would typically be reserved for larger regions

Line 381 - spatial not spatially

Line 391 - note that WorldCover also uses Sentinel-1 imagery, this microwave imagery contributes a different element to the detection with texture and volume content

Line 394 – note that the ESRI map is generated once per year but the Dynamic World updated every 2-5 days and also has per pixel probabilities – were these considered by the authors at all?

Line 411 – remove possess

Line 448 – the definition of grassland is very broad being up to 75% tree cover, with widely varying C stocks between the two extreme ends of the range, so how valuable is it to have such a broad range if the end goal is C calculations, how would the same authors define wooded areas – I would guess the lower range would be less than 75% tree cover and so a grey area is formed which falls into two categories – if the land cover is to be determined by RS then anything covering more than 50% of a pixel is likely to be the dominant class - some discussion of these points is needed

Line 456-9 – as a proportion of land area how well distributed are the number of grids?

Line 459 - as part of the summary, rather than just basing values on global estimates, how do the different statistics vary nationally/regionally/continentally i.e. are the biggest differences in the more remote locations, or where there are a greater range of different grassland types, or where there are certain specific categories of grassland that result in more mis-classifications – there can be some very accurate national land cover maps especially in Europe and this kind of comparison might be useful for individual governments who are tasked with challenges such as carbon reporting

In Table 1, include the values from the three products used throughout the paper for comparison

Figure 1 shows large areas of grasslands with no sites within the study – if multiple reviewers were to assess some of these areas a sense of the land cover there could be gained – also the caption says MOST sites contain 11 grids, but how many do not and why not

Figure 2 – refer to pixels not images throughout the caption

Supplementary material

Line 22 - explain a bit more about how the 504 sites chosen – were they entirely based on the individual contributors' knowledge? If yes then how was representation between the different grassland types ensured, and were there any standardised comparisons to assess the reliability of the different contributors e.g. this example from the world of glaciology https://link.springer.com/chapter/10.1007/978-3-540-79818-7_7

Line 76-77 - bear in mind that the QGIS Bing and Google maps are compiled from different dates and not necessarily from 2020, this can't be avoided but needs to be mentioned as a possible source of discrepancy

Line 220 – were

Figure S1 – why not show the ESRI land cover product from 2020 for direct comparison?

Table S1 - show all data to 5 decimal places for consistency

Table S3 - all formatting should be the same i.e. central or right justified

Table S5 – what does a GEE scale of 500 mean in terms of m/km, and why not calculate values at their native resolution - also might be better to show the areas to '000km²

Version 1:

Decision Letter:

18th September 2025

Dear Dr. MacDougall,

Thank you for your patience while your revised manuscript "The global extent of the grassland biome and implications for the terrestrial carbon sink" (NATECOLEVOL-25030649A) was under review. On the basis of the feedback received, we will be happy in principle to publish the manuscript in Nature Ecology & Evolution, pending minor revisions to satisfy the reviewers' final requests and to comply with our editorial and formatting guidelines.

[redacted]

Reviewer 1's comments

The authors did a very thorough revision to address my concern. I recommend publication.

[Editor's note: Reviewer 1 also addressed the responses to Reviewer 2, who was not available]

Most of the comments provided by Reviewer 2 pertain to clarifications or are just simply suggestions for improvement. A few key comments that warrant attention though:

"The abstract would benefit from a total rewrite with an emphasis on the discrepancies and challenges of the existing grassland maps, and the new estimate of grassland cover"

The authors have updated both the title and the abstract to address issues of discrepancies and challenges of mapping.

* * *

"Line 252, why are the challenges particularly pronounced for grasslands?"

The authors have added an illustrative figure (F2) with text that clarifies the matter.

* * *

"Line 373, 379 and elsewhere, there is inconsistency in the use of images, areas, cells and pixels - I recommend removing all references to the individual cells as images or areas as that would typically be reserved for larger regions"

I believe this was addressed; "pixels" are now used throughout the manuscript

* * *

"Line 448 – the definition of grassland is very broad being up to 75% tree cover, with widely varying C stocks between the two extreme ends of the range, so how valuable is it to have such a broad range if the end goal is C calculations, how would the same authors define wooded areas – I would guess the lower range would be less than 75% tree cover and so a grey area is formed which falls into two categories – if the land cover is to be determined by RS then anything covering more than 50% of a pixel is likely to be the dominant class - some discussion of these points is needed"

This is an important point that I'm glad the reviewer noted (I should have picked up on this myself). The authors have certainly addressed the problem by providing more details and information on the issue of definition. Don't know if reviewer 2 would be satisfied with the added details but the issue is addressed in my opinion.

* * *

"Line 22 - explain a bit more about how the 504 sites chosen – were they entirely based on the individual contributors' knowledge? If yes then how was representation between the different grassland types ensured, and were there any standardised comparisons to assess the reliability of the different contributors"

I commented on this myself; the authors now explain how they reached that number.

* * *

In summary, the authors have been diligent in addressing the comments and concerns of both reviewers. It is my opinion that Reviewer 2 did not identify any major issues (it's mostly about clarifications and suggestions for improvement) and that the her/his comments have been adequately addressed. I recommend publication.

Reviewer #1 (Remarks to the Author):

While the implications of using different maps on estimates of carbon stocks warrant investigation, the manuscript suffers from shortcomings that need to be addressed before publication can be considered.

We are thankful for the reviewers' work and positive assessment of our manuscript and appreciate the constructive feedback.

First, the manuscript is not aligned with the remote sensing literature that have discussed issues of accuracy, bias, and uncertainty for decades. The notion that classification "errors ... must be rectified" if remote sensing-based products are to be used for decision-making, for reporting areas, in subsequent analysis, etc., has been emphasized by the remote sensing community for decades. Multiple articles on scientific inference versus pretty satellite images, issues of sampling techniques, and protocols for unbiased estimation of areas, have been published in the remote sensing literature; the manuscript in question is completely disconnected from this wealth of literature. (Much of this literature has been published in the journal *Remote Sensing of Environment* and it is telling that the manuscript in question does not cite a single article published in that journal.) The authors state that their "work differs [because] images are validated by experts with direct on-the-ground familiarity of their sites" – again, it is strongly recommended by the RS community that not only have land cover experts collect reference observations but to estimate reference data uncertainty and interpreter variability.

Yes – no disagreement with any of this. Two of our authors in particular – Siewert and Sulik – are remote-sensing research leaders and are well versed in these issues. We were aware of this literature, and familiar with the long-standing acknowledgement of challenges re: field validation by the remote-sensing community. We have worked to fix this, especially in the Methods. We did not mean to imply that we were reinventing the wheel by ignoring past work on this topic, so we have better matched our paper with the existing literature. The source of this oversight was somewhat intentional initially, as we made a conscious choice in framing the story towards an ecological and policy-based audience with a focus on the question of global grassland extent and consequences for estimates of the terrestrial C sink. In our revised version, we have worked to better connect our efforts to past work. Several new references have been added to highlight previous work evaluating land cover products, including from the journal *Remote Sensing of Environment* as suggested by the reviewer. We have aligned several of the terms used in the original manuscript with remote sensing specific terms, e.g. using Producers'/Users' Accuracy. We have also revisited the literature regarding validation efforts and expertise and remain convinced that there is space, and indeed a need, for improved protocols and closer collaboration with field scientists. This will be even more relevant in the future, as more fine-scaled remotely sensed land cover products are developed.

Second, if the authors aim to provide a global unbiased sampling-based estimate of the area of grassland (and estimates of commission and omission error), the reference sample data must be collected under a probability design with attention to the inclusion probabilities if using traditional design-based estimators (with the estimators corresponding to the sampling design). If the sample data were not collected under a probability design, it is necessary to invoke a model-based inference framework. Yes, our sampling was not purely random so used an inference-based approach to estimate an adjusted/corrected total for global grassland area – this was previously under-described.

But the manuscript provides no discussion of sampling designs, construction of estimators, or of inference frameworks. While there's a long description of "Generation of ground truth grids," it's unclear if this approach conforms to a probability design.

We fully agree – we mentioned none of this in the original submission. We have added considerable discussion of these issues in the main text and the Methods in the Supplementary section, all of which catalogue potential deviations from a “probability sampling protocol” and how we worked to address them.

First, we now discuss this issue in the context of our sampling design. For example, the determination of inclusion probabilities generally requires that the selection of sites (the samples) be truly randomized. We took a number of steps to reduce bias in site selection, but we could not fully randomize our choices. Full and true randomization would require locating sites in all biomes (not just grassland) and then subsequently finding PIs for the sites that were identified. We had to do the opposite – find PIs and then use whatever grassland sites they were affiliated with. This created some geographical bias as we discuss. We worked exhaustively to find under-represented sites and were able to successfully gap-fill. And yet, there are still some areas we missed (see Figure 1). However, we did not select sites with any prejudice to grassland type and our widening of grid selection to a 78 km² area around each core site helped increase the types of cover we analysed (e.g., 27% of our pixels are non-grassland features [~101,000 of 384,600 sampling units]). We conducted nearest-neighbour analysis to test for site clustering (we found none, although our sites were more “dispersed” than random). We plotted the spread of our sites within the major bioclimatic zones of the Earth defined by temperature and precipitation (Fig. S5) – again, we see no bias. We took multiple steps to account for data errors (e.g., Stehman 2001), which can be a source of bias in many studies like ours (e.g., missing images, cloud cover, incorrect image labelling) – we now go into greater detail now how with addressed all of these potential sources of data error. We tested for “strata” bias (Olofsson et al. 2014) using residual analysis associated with a range of biophysical factors, examining whether the sensitivity of LCLU estimates varies by slope, NDVI etc. (Fig. S3) – there was small but inconsistent trends. Finally, we sought to create a reference classification that was as large as some of the so-called “manually validated” non-expert datasets used by the European Space Agency, ESRI, and Google (e.g., to reduce error by elevating sample effort). For example (and as we discuss in the paper) one of the ESA's manually validated datasets has 141,000 images for ALL biomes globally, versus our 384,600 manually validated images for one biome covering ~23% of the earth's terrestrial surface. None of these issues can replace truly randomized sampling, but in combination they help address potential bias.

Second, we adjusted our calculations of estimated/adjusted grassland error based on these considerations. In our original submission, we used a crude “back of the envelope” approach to estimate the most likely true estimate of grassland occurrence. We chose this because it was easy to describe, based on our measures of over- and under-estimation error. But the risk, as implied by the reviewer, is that this estimate might be open to one or a range of hidden biases arising from issues including our non-random selection of sites. This is a valid point. To address this, we consulted the literature for solutions, including some highly cited recommendations that discuss estimation errors in land cover datasets:

1. Olofsson, P. et al., 2014 *Remote Sensing of Environment* 148, 42–57.
2. Olofsson, P. et al., 2020 *Remote Sens. Environ.* 236, 111492.
3. Stehman, S. V. (2001). *Photogrammetric Engineering and Remote Sensing*, 67, 727–734.
4. Tyukavina, A. et al., 2025 *Remote Sens. Environ.* 324, 114714.

5. Stehman, S. V. & Foody, G. M., 2019 *Remote Sens. Environ.* 231, 111199.
6. Foody, G. M. 2002. *Remote Sens. Environ.* 80, 185–201.
7. Strahler, A. et al. 2006. EUR 22156 EN — DG. Luxembourg: Office for Official Publications of the European Communities (48 pp.).

In short, we used a modified approach based on the Olofsson papers especially the 2014 paper (“Good practices for estimating area and assessing accuracy...” *Remote Sensing and Environment*). Foundationally, we meet most of the criteria needed for area estimation given that we possess the overall accuracy by each LCLU product, their user's accuracy (or commission error), their producer's accuracy (or omission error), as well as calculating the area of grassland globally for each (Table 1). We use these measures to infer adjusted global grassland totals for the three LULC products, based on a “proportionality” formula now described in the Methods. We acknowledge that a more sophisticated form of analysis could test for bias risks as well (such as Bayesian modelling), but these lack the elegant simplicity and clarity of our adjusted approach. We also think it is unlikely that our outcomes would change, in terms of creating an estimated final distribution for the grassland biome. There are also challenges in fully replicating the Olofsson model, given our described non-random grassland-specific approach to finding our 504 sites – the Olofsson approach is structured around random site locations as part of their “probability sampling design”. As stated, randomly locating our sites would not work because grasslands themselves are not randomly distributed globally. We now discuss many of these issues in the “caveats” paragraph in the main paper (third last paragraph) and in the Methods section. Also, in alignment with Olofsson et al. 2014, we now frame our data as a “reference classification” (their term) which is critical for accuracy assessment of pixels.

Reviewer #2 (Remarks to the Author):
Replies in bold font.

This is a very interesting paper that addresses a common issue affecting global land cover maps - how accurate are they, and how can the most accurate one be determined, and as the authors highlight, in the absence of being able to answer those questions there is a tendency to quote what someone else has written because that has already been accepted by peer review as being correct!

The paper is well written, and synthesises a large amount of information succinctly - clearly a lot of time and effort was invested in trying to gather the individual experts and generate the image files for them to ensure global coverage in so far as possible. Most of the comments below are for clarification and discussion rather than expecting further work to be conducted.

We would like to thank the reviewer for their work and constructive criticism, and are pleased about their positive judgement of our work.

Turning first to the main paper.

The abstract lacks detail on the results of comparing the land cover maps, and focuses instead on the carbon storage, which is not the focus of the paper or its title. The abstract would benefit from a total rewrite with an emphasis on the discrepancies and challenges of the existing grassland maps, and the new estimate of grassland cover - especially if the authors want to be quoted as the new Loveland et al (2000) authoritative estimate of grassland extent.

Thank you for this comment. Our paper had two main objectives - to assess the accuracy of grassland mapping given the large discrepancy in current estimates, and to explore the implications of these mapping challenges for how grasslands contribute to the size of the terrestrial C sink. As the reviewer correctly pointed out, our former title only dealt with the first issue while our former abstract mostly dealt with the latter. We have worked to fix both problems. We have changed both the title and the abstract, including adding detail (within the text limits) on discrepancies and challenges of mapping. And yes, we hope that once published our paper would indeed become an authoritative source for cited estimates on grassland coverage - both White and Loveland have been cited 1000s of times, and continue to be cited widely, even though they are both badly outdated (as we now explain in the updated Discussion – both relied on satellite technology that was cutting edge in 2000 but is no longer suitable).

The main text needs some additional clarification and discussion in places as noted below.

Around line 240 - although the three products are discussed in detail in the Supplementary Materials section, it would be helpful to have a few extra comments on the nature of the products given that they form the basis of the paper, e.g. years of availability, frequency of update, method of generation etc.

Yes, this makes sense. Arguably the biggest issue, now discussed in much more detail in the main paper and in the supplementary Methods section, is that the performance of the three LULC products differs profoundly even though they largely derive from similar data sources (Sentinel satellite imagery). As such, it is the choices that the product developers make – centered on how they define and classify grassland - that dictates product accuracy (or lack thereof). We did not want to provide too much technical detail in the Introduction and Discussion of the main text as it can get nuanced and “listy” but have nonetheless increased description of the nature of the products. It is in the supplementary section where we drill down with more information, e.g., discussing how all three LCLU products are derived using Sentinel-2 satellite imagery, with ESA also using Sentinel 1 data. We describe how LC and DW share the same training dataset and are based on deep learning models, while WC is based on its own training data and random forest modelling. We also point out how DW is updated in near-real-time with new sentinel imagery, while LC is intended to be updated annually, while WC was meant to be a standalone 2020 map, but got an updated version in 2021. And so on. These are important details that help clarify from where and how the mapping discrepancies arise.

Line 252, why are the challenges particularly pronounced for grasslands?

This is the topic sentence (now starting at line 243) of the paragraph that thoroughly discusses this issue. No changes were made. As discussed in the cited Venter et al paper, grasslands have variable and sometimes confusing spectral signals due to a range of factors described in the text. Grasslands also tend to have unusually “fuzzy” boundaries, especially when open grassland transitions to wooded grassland savanna – the on-the-ground researcher can easily tell that the savanna remains “grassland” by definition (because the understory remains grass-dominated) but the satellite cannot because the canopy hides the understory from view. This issue was one motivating factor for the creation of Figure S2, where we show four visual examples of grassland that would be labeled “forest” from space.

Wooded grassland: Birch forest tundra
grassland (Scandinavia)

Wooded grassland: oak
savanna grassland (central US)

Wooded grassland:
tropical forested savanna (Thailand)

Wooded grassland:
Acacia grassland
(Africa, Australia)

Line 258, close the brackets

Done

Line 272, MODIS is a sensor on the TERRA and AQUA satellites

Changed to "...of the Moderate Resolution Imaging Spectroradiometer (MODIS satellite) on board the TERRA and AQUA satellite constellation."

Line 339, one other cause of the discrepancy is the spatial resolution of the datasets cited in the 2000 papers e.g. Loveland et al use AVHRR 1km data which is invariably going to result in aggregation of smaller units - modifiable areal unit problem scale effect

Great comment. Indeed, this entire section has been rewritten and clarified. Specific to this comment, we have now described how both White et al. 2000 and Loveland et al. 2000 are derived from relatively out-dated "Advanced Very-High-Resolution Radiometer" (AVHRR) satellite data with only a 1 km resolution. We also go into more detail on why these papers make such large (but error-inflated) estimates based on how they define grasslands.

FYI, our intention is not to unfairly criticize these papers – indeed they are groundbreaking which is why they have been cited 1000s of times. We simply want to use them to illustrate how errors can arise. Indeed, we would love our paper to "replace" these papers as the default citation for grassland extent. For example, the leading paper on grassland C sequestration in the last five years is Bai & Cotrufo 2022 with almost 900 citations (*Science* 377: 603-608). Their paper literally begins with this now-erroneous statement "Grassland ecosystems cover an area of 53.2 million km² accounting for ~40.5% of the earth's land surface..." This is a CLASSIC example of the "scientific telephone effect" because they are relying solely on White et al. 2000 for their numbers on grassland extent. Even White et al. never meant their estimates to be taken literally – they intentionally called their paper a "pilot" study to emphasize that it was a coarse and almost "back of envelope" approximation.

Line 373, 379 and elsewhere, there is inconsistency in the use of images, areas, cells and pixels - I recommend removing all references to the individual cells as images or areas as that would typically be reserved for larger regions

Yes. We have revised the way we refer to our sampling units and are now using the word pixel throughout the manuscript.

Line 381 - spatial not spatially

Corrected

Line 391 - note that WorldCover also uses Sentinel-1 imagery, this microwave imagery contributes a different element to the detection with texture and volume content

This information was indeed missing and has now been added.

Line 394 – note that the ESRI map is generated once per year but the Dynamic World updated every 2-5 days and also has per pixel probabilities – were these considered by the authors at all?

The information about the update frequency is now added in the manuscript. We worked with a snapshot of the data for 2020-2021 because those are the data used by World Cover (it's not updated regularly, unlike LC and DW). We avoided accounting for temporal changes because of this. That being said, we discuss the potential effect of temporal deviations at the end of the supplement information re: potential sources of error.

Line 411 – remove possess

Done

Line 448 – the definition of grassland is very broad being up to 75% tree cover, with widely varying C stocks between the two extreme ends of the range, so how valuable is it to have such a broad range if the end goal is C calculations, how would the same authors define wooded areas – I would guess the lower range would be less than 75% tree cover and so a grey area is formed which falls into two categories – if the land cover is to be determined by RS then anything covering more than 50% of a pixel is likely to be the dominant class - some discussion of these points is needed

Great point. We stand by our definition (now added to the Introduction of the paper, to make it clear how we built our “reference classification”) but needed far more detail on this issue. The key issue here is the understory, rather than the relative canopy cover of woody plants: is the understory dominated by grassland taxa or not? 75% is a relatively arbitrary approximation that is generally associated with the upper threshold of canopy shade under which grassland understory taxa can persist. We have worked to clarify this issue, focusing in particular on greater description of our data: our data are targeting the presence of grassland ground taxa, but NOT the % of tree cover. This stands in sharp contrast to remote sensed images, which simply cannot distinguish understory cover once tree cover increases. This is why products such as World Cover restrict their definition of grassland to canopy cover <10% per spatial unit.

There are two major implications here.

First, we have a much broader classification of grassland than is typical (up to 75% woody plant cover) and yet we see MUCH less grassland globally than is usually reported. This is because the most widely cited products such as Loveland and White classify “shrubland” as grassland, thereby significantly and erroneously conflating grassland estimates because not all shrublands support grassland taxa (as our data show). We now provide much more detail on this issue in the main text, as well as the Methods.

Second, despite our definition being broad and indeed likely capturing a wide range in C stocks given how C cycling can differ widely by grassland type, our C calculations stand to be more accurate because our per area estimates are more realistic. The state-of-the-art 2023 UN FAO (Dondini et al.) report seeks to tackle estimates of grassland C, but uses 2010 land cover data that we show to significantly UNDER-estimate global C stocks.

These two major implications are central novel findings of our paper, so we have edited our text to make these clearer.

Line 456-9 – as a proportion of land area how well distributed are the number of grids?

We have addressed this by adding the following analysis described in the third last paragraph in the main text and in the supplemental methods under Spatial Distribution of Site Locations Section.

Line 459 - as part of the summary, rather than just basing values on global estimates, how do the different statistics vary nationally/regionally/continentally i.e. are the biggest differences in the more remote locations, or where there are a greater range of different grassland types, or where there are certain specific categories of grassland that result in more mis-classifications – there can be some very accurate national land cover maps especially in Europe and this kind of comparison might be useful for individual governments who are tasked with challenges such as carbon reporting

We have now clarified/expanded our analysis on potential sources of bias, including by continent. By major climate drivers (Fig. S5) and other biophysical features such as NDVI (Fig. S3).

In Table 1, include the values from the three products used throughout the paper for comparison. These values have now been added.

Figure 1 shows large areas of grasslands with no sites within the study – if multiple reviewers were to assess some of these areas a sense of the land cover there could be gained – also the caption says MOST sites contain 11 grids, but how many do not and why not

Yes, our data come from 500+ grasslands globally, but still we have missed some areas. Our data are thus a large sample, rather than being fully representative. Sadly, we spent over a year proactively but sometimes unsuccessfully reaching out to grassland scientists to participate, especially in under-represented areas such as northern Africa (Tunisia, Morocco, Egypt) and central Asia (Kazakhstan, Uzbekistan). On the other hand, we were able to get sites in areas that are typically absent from global grassland databases (e.g., the Guyanese shield, northern Russia, Iran, Mongolia). Based on sampling probability, we trust that the large size of our database at three spatial scales (site, grid, cell) similarly represents the relative proportion of land cover types found globally in grasslands.

Figure 2 – refer to pixels not images throughout the caption

This has been revised and fixed for the entire manuscript.

Supplementary material

Line 22 - explain a bit more about how the 504 sites chosen – were they entirely based on the individual contributors' knowledge? If yes then how was representation between the different grassland types ensured, and were there any standardised comparisons to assess the reliability of the different contributors e.g. this example from the world of glaciology https://link.springer.com/chapter/10.1007/978-3-540-79818-7_7

We added the following explanatory statement: “The primary focus of the outreach was to create an expansive distribution of locations across the grassland biome. We did not emphasize grassland type (e.g., tundra, savanna, open grassland) nor any other type of attribute (e.g., grazed/ungrazed, invaded by plants or uninvaded)) when selecting our sites - the only requirement was that they be some form of grassland.”

We also added the following section in the Supplement under “Satellite Basemap and Validation Limitations”: “For our analysis, we did not cross-validate among researchers. In some contexts, this might introduce subjective variability in the overall quality of the results. However, we found the opposite: local expertise of researchers was irreplaceable for determining the difference between a grassland and non-grassland land cover units, and to identify and label each pixel as one of our 16 different cover types. In some cases when needed, our core team responded to requests for clarification via email or video call to help ensure consistency regarding the class definitions. We also conducted informal trials where several of our participant grassland experts scored pixels from regions where they lacked familiarity – the error was typically very high (i.e., high interpreter uncertainty¹⁴). Even for sites that were clearly open grassland lacking tree and shrub cover, the lack of local site familiarity made it impossible for grassland experts from elsewhere to reliably tell if these areas were natural grassland (classified as “1” – see Table S2) or planted pasture (classified as “5”). For areas with tree or shrub cover or for seasonally flooded grassland, it was even harder to correctly and consistently make the proper classification (Fig. S1). This exercise once again highlights the critical need for local expertise when ground-truthing satellite imagery.”

Line 76-77 - bear in mind that the QGIS Bing and Google maps are compiled from different dates and not necessarily from 2020, this can't be avoided but needs to be mentioned as a possible source of discrepancy

This is now mentioned in the section on limitations of the methods at the end of the Additional Methods.

Line 220 – were

Fixed

Figure S1 – why not show the ESRI land cover product from 2020 for direct comparison?

Done

Table S1 - show all data to 5 decimal places for consistency

All data is now shown to 5 decimal places. Furthermore, the table has been reformatted to short format.

Table S3 - all formatting should be the same i.e. central or right justified

Table S3 has been updated to accommodate the reviewers comment.

Table S5 – what does a GEE scale of 500 mean in terms of m/km, and why not calculate values at their native resolution - also might be better to show the areas to '000km²

We now specify that this as 500m as in 500 m x 500 m resolution of pixels and changed the wording to be less confusing. All values are now rounded to 000km².